# Functional information from clinically-derived drug resistant forms of the Candida glabrata Pdr1 transcription factor

**Lucia Simonicova**, **W. Scott Moye-Rowley** *

Department of Molecular Physiology and Biophysics, Carver College of Medicine, University of Iowa, Iowa City, IA, United States of America

* scott-moye-rowley@uiowa.edu

**Data Availability Statement:** All relevant data are within the manuscript and its Supporting Information files.

## Abstract

Azole drugs are the most frequently used antifungal agents. The pathogenic yeast *Candida glabrata* acquires resistance to azole drugs via single amino acid substitution mutations eliciting a gain-of-function (GOF) hyperactive phenotype in the Pdr1 transcription factor. These GOF mutants constitutively drive high transcription of target genes such as the ATP-binding cassette transporter-encoding *CDR1* locus. Previous characterization of Pdr1 has demonstrated that this factor is negatively controlled by the action of a central regulatory domain (CRD) of ~700 amino acids, in which GOF mutations are often found. Our earlier experiments demonstrated that a Pdr1 derivative in which the CRD was deleted gave rise to a transcriptional regulator that could not be maintained as the sole copy of *PDR1* in the cell owing to its toxically high activity. Using a set of GOF *PDR1* alleles from azole-resistant clinical isolates, we have analyzed the mechanisms acting to repress Pdr1 transcriptional activity. Our data support the view that Pdr1-dependent transactivation is mediated by a complex network of transcriptional coactivators interacting with the extreme C-terminal part of Pdr1. These coactivators include but are not limited to the Mediator component Med15A. Activity of this C-terminal domain is controlled by the CRD and requires multiple regions across the C-terminus for normal function. We also provide genetic evidence for an element within the transactivation domain that mediates the interaction of Pdr1 with coactivators on one hand while restricting Pdr1 activity on the other hand. These data indicate that GOF mutations in *PDR1* block nonidentical negative inputs that would otherwise restrain Pdr1 transcriptional activation. The strong C-terminal transactivation domain of Pdr1 uses multiple different protein regions to recruit coactivators.

## Author summary

Resistance to antibiotics is a major threat to the continued use of these lifesaving chemotherapeutic drugs. This problem is especially acute in the case of antifungal drugs as only 3 classes of these compounds exist. The pathogenic yeast *Candida glabrata* acquires resistance to the azole class of antifungal drugs by developing hyperactive alleles of the *PDR1*

**Funding:** WSMR received grants NIGMS 49825 and NIAID 152494 from the National Institutes of Health (https://www.nigms.nih.gov/ and https://www.niaid.nih.gov/). The funders had no role in study design, data collection and analysis, decision to publish, or preparation of the manuscript.

**Competing interests:** The authors have declared that no competing interests exist.

gene, encoding a major inducer of azole resistance. We provide evidence that these hyperactive mutant proteins identify different negative inputs that would otherwise repress the transcriptional activity of Pdr1. Mutational analysis of the extreme C-terminus of Pdr1 indicated that this region exhibited multiple different interactions with coactivator proteins required for normal transcriptional activation of target gene expression. The data reported here shed light on the complicated nature of regulation of Pdr1 activity and identify domains in this protein that are bifunctional in their role to ensure normal factor activity. A detailed understanding of the molecular control of Pdr1 will allow strategies to be devised to reverse the azole resistance triggered by mutant forms of this protein.

## Introduction

Antifungal drugs are limited to three classes of compounds that are routinely used in the clinic [1]. The most commonly prescribed of these are the azole drugs with fluconazole being the most typically used of these antifungal agents [2]. Fluconazole, like all azole drugs, acts to inhibit lanosterol α-14 demethylase, a key enzyme involved in ergosterol biosynthesis (Discussed in [3]). The pathogenic yeast *Candida glabrata* is increasingly associated with candidemia, likely due in part to the ease with which this organism develops resistance to fluconazole and other azole drugs [4–6]. The primary resistance mechanism in *C. glabrata* occurs via acquisition of single amino acid substitution mutations in a transcriptional regulator called Pdr1 (reviewed in [7]). These gain-of-function (GOF) forms of Pdr1 behave as hyperactive inducers of transcription and elicit high-level fluconazole resistance via overexpression of target genes such as the ATP-binding cassette (ABC) transporter-encoding gene *CDR1* (Recently reviewed in [8]).

While the consequences of GOF forms of Pdr1 are well-documented, the mechanisms behind these phenotypic changes in function are poorly understood. Structure/function analysis of Pdr1 has provided evidence for this factor having the typical structure of a $Zn_2 Cys_6$ DNA-binding domain (DBD) transcription factor [9,10]. The DBD of this protein is located in the first 300 amino acids of the 1107 residue protein chain. The central regulatory domain (CRD) is ~700 amino acids and links the DBD with the extreme C-terminal transactivation domain (TAD). GOF mutations in *PDR1* cluster to the CRD and TAD regions of this protein [11]. A Pdr1 derivative lacking the CRD is a strong transactivator that is toxic when expressed as the sole source of Pdr1 activity in *C. glabrata*. It is thought to kill the cell via squelching [10], indicating the essential nature of maintaining some degree of the negative regulation normally imposed on this factor (recently reviewed in [12]).

The simplest explanation for the action of GOF mutations in *PDR1* is that these lesions alter protein structure such that a common negative regulatory input is lost. Experiments aimed at identifying interacting proteins with the Pdr1 TAD demonstrated the Mediator subunit Med15A binds to the C-terminal 34 residues and provided evidence that this binding is required to fully drive induced gene expression [13]. Genetic and biochemical experiments have identified two trans-acting negative regulators of Pdr1 activity—Jjj1 and Bre5, although their target region within Pdr1 remains unknown [14,15]. While a broad spectrum of GOF forms of Pdr1 have been identified, how these lesions trigger the high-level transactivation of Pdr1 target genes is not understood.

Here we provide evidence that different GOF alleles clustered either in the CRD or the TAD cause hyperactivation of Pdr1 via different mechanisms. Detailed analyses of the C-terminal TAD indicate that multiple coactivator proteins are likely to be contacted by this

segment of the protein during transcriptional activation. We also demonstrate that a mutation in the TAD has two different effects on Pdr1 function. This mutation (D1082G) represents one of the strongest GOF alleles when assayed in the context of the wild-type factor [10,11]. Surprisingly, when D1082G is introduced into the structure of the toxic Pdr1 derivative lacking the CRD (Δ255–968 Pdr1), this combination mutant is now able to serve as the sole copy of Pdr1 in the cell. When assayed in the Δ255–968 Pdr1 context, D1082G weakens the transactivation capability of this derivative. Our data are consistent with a model in which multiple, different negative inputs act to repress Pdr1 activity via control of the strong TAD that contacts with several coactivator proteins including Med15A. These genetic data reveal the multicomponent negative regulation of Pdr1 as well as the suite of coactivators that this factor engages to drive gene expression.

## Results

### Pdr1 has multiple functional domains

Previous analyses of Pdr1 have defined at least three different functional domains within this transcriptional regulatory protein (Fig 1). The amino terminal 254 residues define a $Zn_2 Cys_6$ zinc cluster-containing DNA-binding domain (DBD) while the extreme carboxy-terminal 138 amino acids specify the major transactivation domain (TAD). These two critical regions are separated by 715 amino acids that encode the central regulatory domain (CRD) of this factor. We previously reported [10] the characterization of a subset of clinically-derived mutant forms of Pdr1 [11] that all led to the production of a hyperactive form of this transcription factor when present as isogenic changes in the *PDR1* gene. These clinical alleles all map to either the central regulatory or transactivation domains of the protein and are gain-of-function (GOF) forms of the protein. Importantly, loss of the central regulatory domain led to the production of a form of Pdr1 that was so active that it was lethal when present as the only copy of this gene in the cell [10].

One complicating feature for the analysis of different alleles of *PDR1* is the autoregulation of this gene [9]. All of the GOF forms of Pdr1 have increased expression of target genes like the ATP-binding cassette (ABC) transporter-encoding *CDR1* but also of *PDR1* itself [16–18]. As a

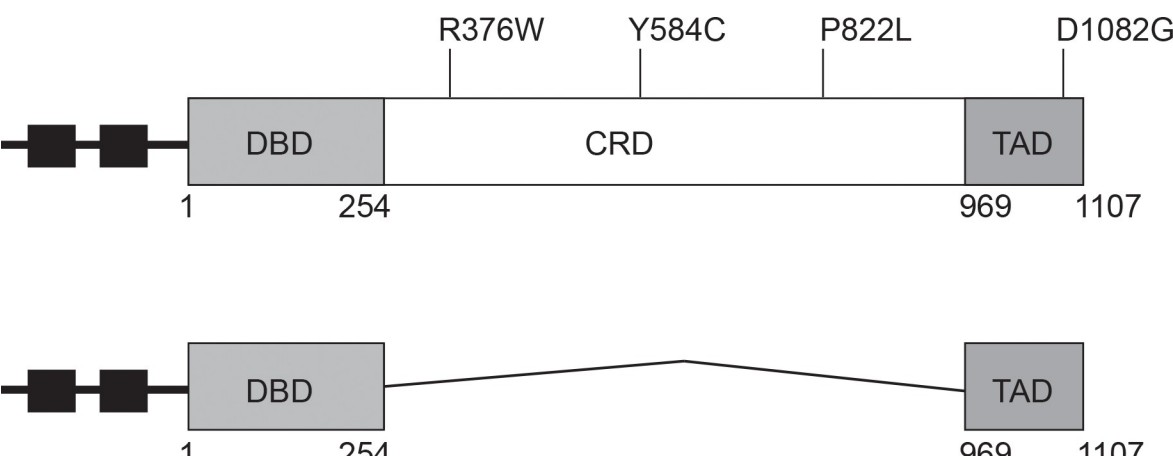

**Fig 1. Graphic representation of Pdr1 gain-of-function mutations used in the study.** Top figure: The full length Pdr1 protein with its functional domains (DBD, DNA binding domain; CRD, central regulatory domain and TAD, transactivation domain) and the location of gain-of-function mutations is indicated. Small square black boxes indicate position of Pdr1 response elements (PDREs) that provide Pdr1 autoregulatory control. Bottom figure: structure of the Δ255–968 Pdr1 is shown.

result of this overproduction, it is difficult to determine if the transcriptional activity of these mutant proteins is actually higher than the wild-type or if these mutants accumulate to a higher level. To determine the cause of the increased expression of Pdr1 target genes in an autoregulation independent manner, we replaced the native *PDR1* promoter with the cognate region from the methionine-repressible *MET3* gene [19]. The resulting *MET3-PDR1* gene fusion allowed production of Pdr1 with no contribution from the autoregulatory circuit that exists in the normal *PDR1* gene.

Low-copy-number plasmids expressing *MET3-PDR1* fusions corresponding to the wild-type *PDR1* gene as well as two different GOF alleles (R376W and D1082G) were introduced into a *pdr1Δ* strain. Representative transformants were then tested for their ability to confer fluconazole resistance in the presence or absence of methionine (Fig 2A).

Expressing wild-type *PDR1* from the *MET3* promoter conferred increased fluconazole resistance compared to the strain with *PDR1* under its native promoter. Both GOF forms of Pdr1 drove higher levels of azole resistance than the wild-type protein irrespective of being under control of the native *PDR1* or the *MET3* promoter, so long as methionine was omitted from the medium. Upon methionine repression, all three forms of *MET3-PDR1* supported lower levels of fluconazole resistance than their corresponding *PDR1* promoter-controlled counterparts due to the inhibition of *MET3* promoter function [10,19]. To examine the correlation between expression of these different species of Pdr1 and azole resistance, we carried out western blot analysis on these same transformants using our previously described Pdr1 antiserum [20].

When produced from the derepressed *MET3* promoter, the wild-type form of Pdr1 accumulated to the highest level compared to either GOF form (Fig 2B and 2C). Importantly, even though produced to roughly a 3-fold excess compared to either GOF form, wild-type Pdr1 supported lower levels of fluconazole resistance and lower levels of Cdr1 expression. This shows that wild-type Pdr1 is still under some negative regulatory input that restricts its transcriptional activity. Since expression from the *MET3* promoter is insensitive to Pdr1 autoregulation, these different forms of Pdr1 are expressed at levels uncoupled from the activity of Pdr1. These data fully support the view that GOF forms of Pdr1 possess a higher specific activity compared to wild type Pdr1 as measured by transcriptional activation of downstream target gene expression.

## Clinically-important GOF Pdr1 forms exhibit non-equivalent behaviour

Having established that at least two different GOF forms of Pdr1 have elevated transcriptional activation, we wanted to examine the basic mechanism(s) underlying how different clinically-derived mutant forms of Pdr1 produce this enhanced target gene expression. As a first step in this analysis, we assessed the genetic dependence of 4 different GOF alleles of *PDR1* on the presence of a component of the transcriptional Mediator complex shown to be required for normal Pdr1 function in *C. glabrata*. The transcriptional Mediator is an interacting complex of ~25 different proteins that act to link activator proteins with RNA polymerase II and consists of 4 different subcomplexes: head, middle, tail and the CDK8/CyclinC module (recently reviewed in [21,22]). Med15A (sometimes called Gal11A) is a subunit of the tail complex, important in Pdr1-mediated transcriptional activation [9,13].

To examine the relative dependence of these Pdr1 GOF forms on Med15A, we introduced low-copy-number plasmids expressing these different *PDR1* alleles into strains lacking the chromosomal *PDR1* locus and either containing or lacking the *MED15A* gene. We analyzed three different GOF alleles that map to the CRD region (R376W, Y584C, P822L) and one that lies within the TAD region (D1082G). We also constructed two combination mutant alleles

**A.**

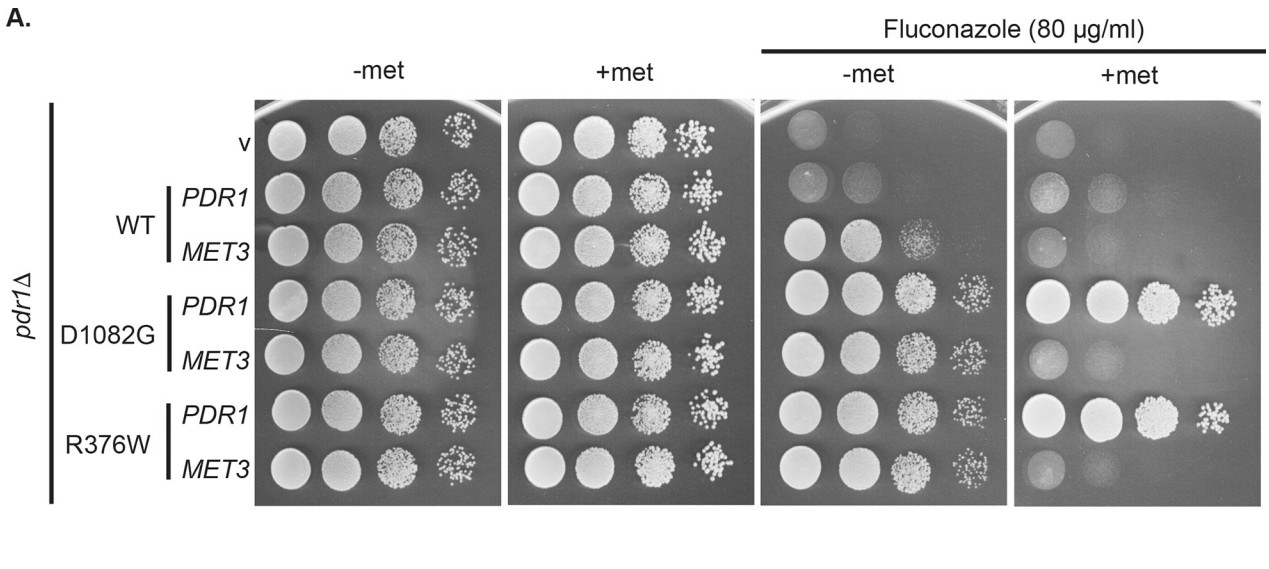

**B.**

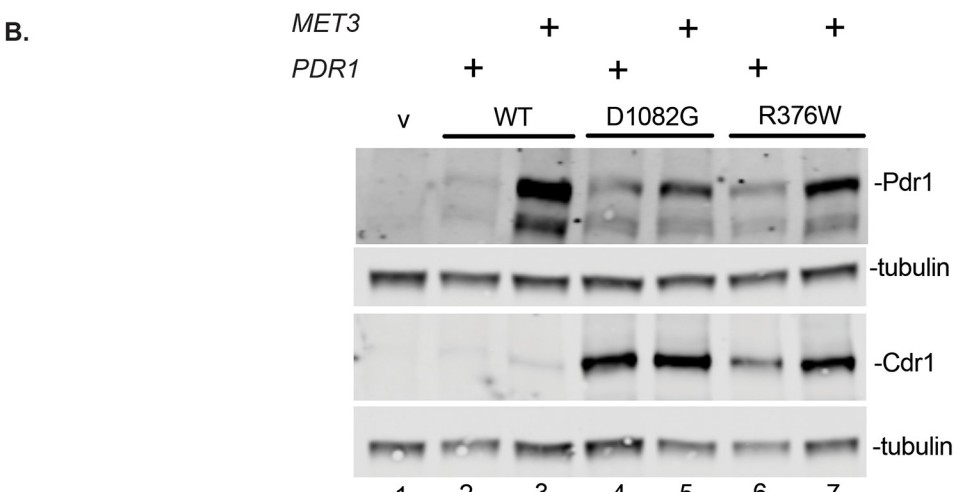

**C.**

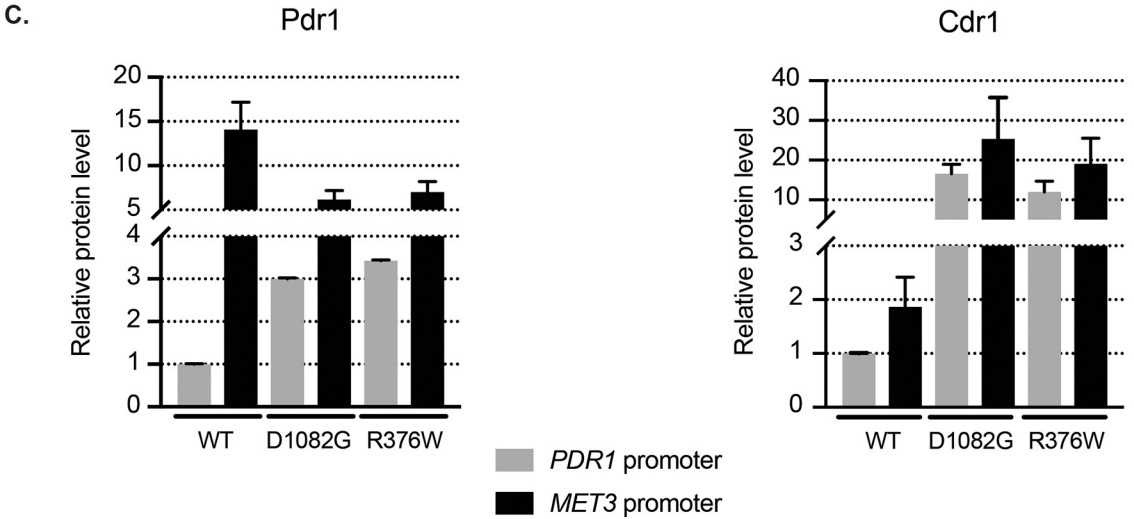

**Fig 2. Pdr1 gain-of-function forms have increased intrinsic capacity for transactivation compared to wild type Pdr1.** A. The *pdr1Δ* strain was transformed with low-copy-number vector (v) or containing *PDR1* wild type or GOF forms D1082G and R376W under control of native promoter (*PDR1*) or the methionine repressible *MET3* promoter (*MET3*). Transformants were grown to mid-log phase in minimal selective media without methionine and aliquot of cells was spotted on minimal media with (2 mM) or without methionine. Fluconazole was present where indicated at the concentration of 80 μg/ml to test for drug resistance. B. Mid-log cells expressing Pdr1 forms from the *PDR1* or *MET3* promoter grown in minimal media without methionine were subjected to protein extraction and analyzed for Pdr1 and Cdr1 levels by western blotting using anti-Pdr1 or anti-Cdr1 antibodies. Tubulin was used as loading control. A representative western blot is shown. C. Relative Pdr1 and Cdr1 protein level among tested strains expressing Pdr1 forms driven either from *PDR1* promoter (grey bars) or *MET3* promoter (black bars) compared to the strain with wild type Pdr1 expressed from *PDR1* promoter. Error bars represent standard error of the mean. All quantitative values for this and all figures are provided as S1 Table (S1 Table).

called either Triple (contains all CRD mutations) or Quadruple (contains all CRD mutations and the TAD substitution). We were interested to know if the mode of action of these mutations were similar and wanted to determine if a single mutant could be influenced by combining it with other, different alleles. Increased function upon the cumulative addition of these mutations would be consistent with different mechanisms being disrupted by these different mutations or with each mutant partially relieving a common regulatory input. Mutants were tested for their ability to confer fluconazole resistance, for the expression profiles of Pdr1 target genes involved in fluconazole efflux and for the steady-state levels of Pdr1 and Cdr1.

The D1082G form of Pdr1 showed the least dependence on Med15A of all the forms of single mutant *PDR1* genes tested as D1082G Pdr1 was able to grow at 10 μg/ml fluconazole in a manner that was nearly Med15A independent (Fig 3A). All single mutants showed strong reduction in growth at 40 μg/ml fluconazole when Med15A was removed from the strain. Importantly, the Triple mutant showed almost complete Med15A independence at 10 μg/ml and retained the strongest level of resistance at 40 μg/ml in the absence of Med15A. This behavior was not shared by the Quadruple mutant as it showed less fluconazole resistance than the D1082G Pdr1 in the absence of Med15A at 10 μg/ml and was very similar in growth to the single mutants at 40 μg/ml. Loss of Med15A revealed that the Triple mutant is the strongest GOF allele, likely due to additive effects of each individual mutation, with the least Med15A dependence while introduction of the D1082G allele into the Triple mutant (to form the Quadruple) actually attenuated the function of the Triple. This was the first indication that the behavior of the D1082G allele was not strictly as a GOF form of Pdr1.

To study the effect of various *PDR1* mutations on expression of plasma membrane transporters involved in azole resistance, we carried out real time qPCR. Besides *CDR1*, expression levels of *CDR2*, *SNQ2* and *YBT1* were assessed in both *pdr1Δ* and *med15AΔ pdr1Δ* genetic backgrounds (Fig 3B). We observed that all mutants showed the most prominent increases in *CDR1* expression compared to the other genes, making *CDR1* the most highly Pdr1 responsive transporter-encoding gene. In comparison to single *PDR1* mutants, the Triple and Quadruple mutants exhibited the highest transcription levels of all four transporter genes. As expected, the expression of the transporter genes in response to these same *PDR1* alleles was notably reduced in all strains in the absence of Med15A. The Triple Pdr1 mutant supported 2-fold higher levels of *CDR1* transcription than the Quadruple and represented the strongest *PDR1* allele in cells lacking Med15A. This is consistent with the observation that the Triple mutant can grow in higher levels of fluconazole in the absence of Med15A compared to all other tested strains.

To evaluate the effect of these mutants on both expression of Pdr1 as well as Cdr1, western blot analyses on wild type Pdr1, R376W, D1082G, Triple and Quadruple were carried out with the appropriate antibodies and quantitated (Fig 3C and 3D). The highest level of expression of Cdr1 was produced by the Triple mutant although this was still reduced upon removal of Med15A. All forms of Pdr1 exhibited expression defects upon loss of this Mediator

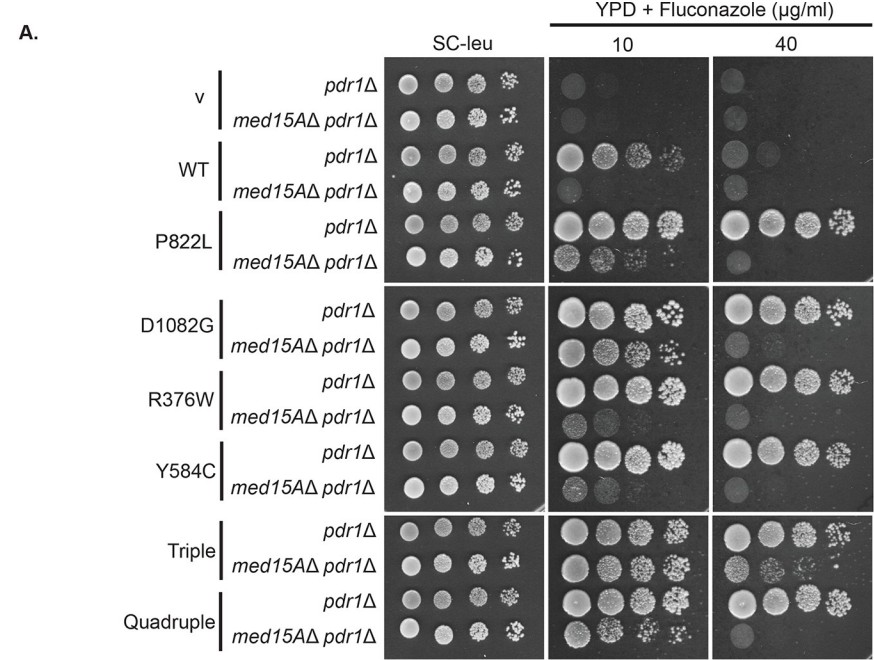

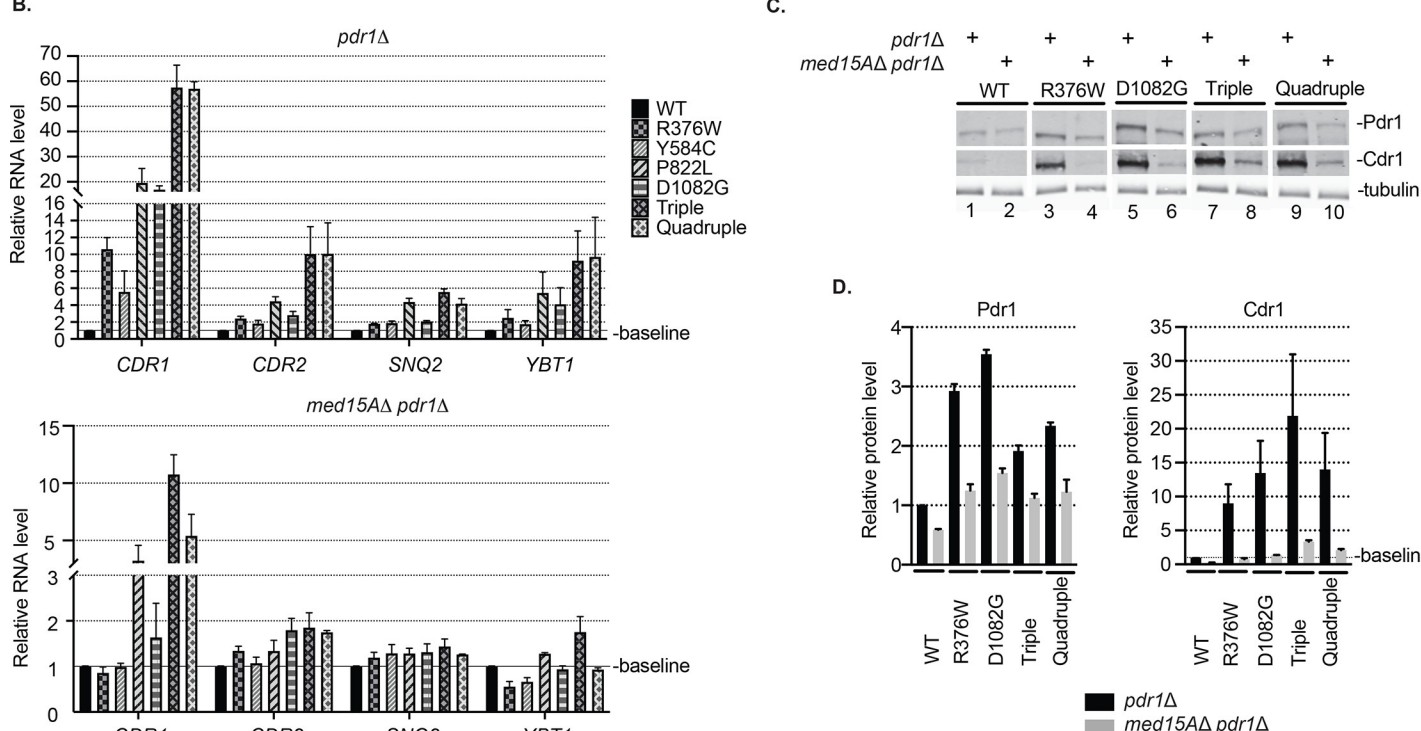

**Fig 3. Various Pdr1 gain-of-function forms exhibit non-equivalent mode of action.** A. Strains *pdr1Δ* and *med15AΔ pdr1Δ* were transformed with low-copy-number vector alone (v) or containing *PDR1* wild type (WT) or various GOF alleles (P822L, D1082G, R376W, Y584C, Triple or Quadruple). Transformants were grown to mid-log phase in liquid minimal selective media and spotted on minimal selective media or YPD media containing fluconazole at the concentration of 10 or 40 μg/ml. B. Mid-log cells expressing various *PDR1* forms in both *pdr1Δ* and *med15AΔ pdr1Δ* genetic backgrounds were grown in minimal selective media, subjected to total RNA extraction, cDNA synthesis and qRT-PCR analysis of the expression of plasma membrane transporters *CDR1*, *CDR2*, *SNQ2* and *YBT1*. Wild type Pdr1 values were used to normalize the expression levels of transporters among strains. The average Ct value for each sample was calculated from the triplicate and normalized to the Ct value of the housekeeping gene *TEF1*. All measurements represent the result of two independent experiments performed on two sets of transformants and the error bars were calculated as standard error of the mean. C. Mid-log cells expressing various *PDR1* forms in both *pdr1Δ* and *med15AΔ pdr1Δ* genetic backgrounds were grown in

minimal selective media, subjected to protein extraction and analyzed for Pdr1 and Cdr1 levels by western blotting using anti-Pdr1 or anti-Cdr1 serum. Tubulin was used as a loading control. A representative western blot is shown. D. Relative Pdr1 and Cdr1 protein levels in *pdr1Δ* and *med15AΔ pdr1Δ* strain transformed with various *PDR1* GOF forms. Levels of Pdr1 and Cdr1 in each strain were normalized to the wild-type protein. Error bars represent standard error of the mean.

component. Levels of Pdr1 were similar across compared mutants, with D1082G form showing the highest expression. As seen for the level of fluconazole resistance supported and expression profiling, the Triple mutant drove the highest level of Cdr1 expression, an effect that was reversed by introduction of the D1082G allele in the Quadruple mutant, although in this assay the effect was less pronounced than when transcription was directly assayed (Compare Fig 3B with Fig 3C and 3D). This was quite surprising since the D1082G *PDR1* allele is one of the strongest GOF mutants we have tested, at least in the context of an otherwise wild-type *PDR1* gene. To probe the basis of this unexpected context-dependent reduction in transcriptional activity shown by D1082G, we introduced this allele into the toxic transactivator Δ255–968 Pdr1 (referred to as internal deletion Pdr1).

## D1082G mutation compromises hyperactivity of internal deletion derivative of Pdr1

We have previously demonstrated that loss of the negatively-acting central regulatory domain (CRD) of Pdr1 led to the production of a hyperactive form of Pdr1 that was unable to be tolerated as the only copy of Pdr1 in the cell [10]. Based on our finding that the D1082G allele appeared to reduce the activity of the Triple mutant, we wondered if this substitution mutation would have a similar effect on the Δ255–968 form of Pdr1. To test this idea, we introduced the D1082G alteration into the amino-terminal TAP-tagged Δ255–968 form of the *PDR1* gene and introduced this new mutant allele into *C. glabrata* cells containing a *pdr1Δ* null allele (Fig 4A). We first compared the efficiency of transformation of the empty vector plasmid as well as this same plasmid containing the wild-type, Δ255–968 or Δ255–968 D1082G forms of the *PDR1* gene in a *pdr1Δ* background (Fig 4B).

Introduction of the D1082G allele into the Δ255–968 form of the *PDR1* gene led to a 2-log increase in transformation efficiency when compared to the same clone containing the wild-type D1082 form as the wild-type TAD caused toxicity in this context [10]. This finding prompted us to characterize the behavior of this altered form of the Δ255–968 Pdr1 in more detail. We introduced the Δ255–968 D1082G Pdr1 form carried on a low-copy-number plasmid into isogenic wild-type and *pdr1Δ* cells and analyzed the ability of this mutant to confer fluconazole resistance, expression of the mutant Pdr1 polypeptide and Cdr1 as above. Comparison with the Δ255–968 form of Pdr1 was only possible in a strain that carried a wild-type *PDR1* gene on the chromosome (Fig 4C).

The presence of the Δ255–968 D1082G Pdr1 in a *pdr1Δ* background led to robust growth at 40 μg/ml fluconazole while the wild-type gene exhibited a strong reduction in growth at this drug concentration. To compare Δ255–968 D1082G and Δ255–968 Pdr1 forms, plasmids containing each of these genes or the empty vector alone were introduced into *PDR1* cells. Both the Δ255–968 forms of Pdr1 drove high level fluconazole resistance, although the Δ255–968 D1082G Pdr1 was reduced compared to Δ255–968 Pdr1 (Fig 4C).

Levels of expression of Cdr1 and the different forms of Pdr1 were compared for the Δ255–968 D1082G and Δ255–968 Pdr1 derivatives (Fig 4D and 4E). In the presence of a wild-type copy of *PDR1*, the Δ255–968 D1082G Pdr1 produced less Cdr1 than did its Δ255–968 counterpart. This correlated with its lower fluconazole resistance above (Fig 4C). The expression of both the Δ255–968 as well as Δ255–968 D1082G form of Pdr1 in the presence of wild type copy of *PDR1* were not significantly different.

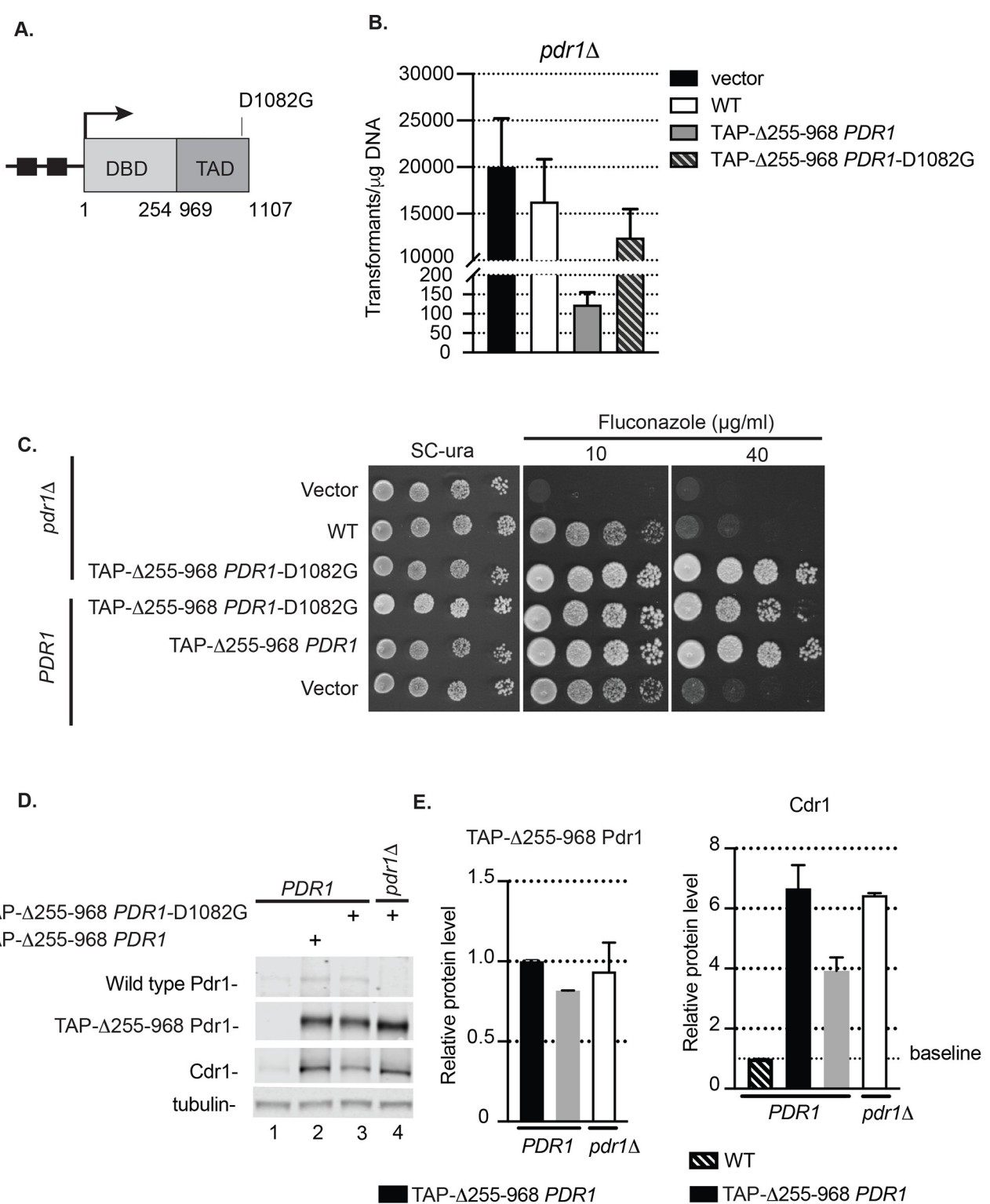

**Fig 4. Insertion of D1082G mutation abolishes the toxicity of *PDR1* lacking the central regulatory domain in *pdr1Δ* cells.** A. Diagram of *PDR1* gene encoding the internal deletion Δ255–968 Pdr1 derivative with D1082G mutation. Functional domains (DBD, DNA binding domain and TAD, transactivation domain) of the protein and relative location of gain-of-function mutation D1082G is indicated. See legend to Fig 1 for further details. B.

Transformation efficiency of vector alone or carrying wild type *PDR1* or internal deletion *PDR1* forms in *pdr1Δ* cells. Note discontinuous ordinate. C. Wild type (*PDR1*) and *pdr1Δ* strain transformed with vector (v) alone or carrying various Pdr1 forms were grown to mid-log phase and tested for fluconazole susceptibility in YPD media containing 10 or 40 μg/ml of drug. D. Mid-log cells expressing internal deletion Pdr1 forms Δ255–968 *PDR1* or Δ255–968 *PDR1*-D1082G in wild type (*PDR1*) or *pdr1Δ* background were grown in minimal selective media, subjected to protein extraction and analyzed for Pdr1, TAP-Δ255–968 Pdr1 and Cdr1 levels by western blotting using anti-Pdr1 or anti-Cdr1 antibodies. Tubulin was used as loading control. A representative western blot is shown. E. Relative TAP-Δ255–968 Pdr1 and Cdr1 protein level in wild type (*PDR1*) or *pdr1Δ* strain expressing various Pdr1 forms. Levels of TAP-Δ255–968 Pdr1 in strains were normalized to wild type strain expressing also TAP-Δ255–968 Pdr1. Levels of Cdr1 were normalized to Cdr1 driven from wild type strain. Error bars represent standard error of the mean.

These data are consistent with the Δ255–968 D1082G Pdr1 being less active in terms of downstream gene transcription than the Δ255–968 Pdr1. However, owing to the lethality caused by the Δ255–968 Pdr1 in *pdr1Δ* cells, direct comparison was difficult. To address this complication, we used an acutely repressible form of wild-type Pdr1 in order to examine function of these two derivatives of Δ255–968 Pdr1 in the same genetic background. We previously showed that the Δ255–968 Pdr1 can be maintained in cells in the presence of the methionine-repressible *MET3-PDR1* fusion gene [10]. The addition of methionine allows depletion of wild-type Pdr1 and acute analysis of the elevated function of Δ255–968 Pdr1.

Low-copy-number plasmids expressing Δ255–968 or the Δ255–968 D1082G Pdr1 derivatives were introduced into *pdr1Δ* cells along with a second plasmid containing or lacking the *MET3-TAP-PDR1* fusion gene. All of these Pdr1 proteins were expressed with an amino-terminal tandem affinity purification (TAP) tag to facilitate immunological detection as described earlier [20]. Transformants were grown to mid-log phase and placed on the indicated media to assay for the ability to grow in the presence of fluconazole (Fig 5A).

These transformants behaved as expected. Methionine repression of *MET3-TAP-PDR1* was lethal in the presence of Δ255–968 Pdr1 but fluconazole hyper-resistance could be seen when the Δ255–968 D1082G Pdr1 protein was expressed in the same cell.

We measured the acute effects of these different fusion genes on expression of Cdr1 to test for differences between the transcriptional activation capability of these different alleles of Δ255–968 *PDR1*. These same transformants were grown to the mid-log phase and then for an additional 6 hours with or without methionine addition. At this point, whole cell protein extracts were prepared and analyzed by western blotting using antibodies against Pdr1 or Cdr1 (Fig 5B and 5C).

After the 6 hour treatment with methionine, the levels of both the internal deletion form of Pdr1 polypeptide and Cdr1 were higher in the presence of Δ255–968 Pdr1 than its isogenic Δ255–968 D1082G Pdr1 derivative (Fig 5B, compare lanes 2 and 6 and Fig 5C, compare Pdr1 and Cdr1 levels in columns 3 and 5). This suggests that the transactivation function of the Δ255–968 Pdr1 protein was more potent than that of the Δ255–968 D1082G Pdr1. Again, this interpretation was complicated by the autoregulatory loop that was retained in both Δ255–968 Pdr1 derivatives. To eliminate this complication, we expressed these two internally deleted Pdr1 proteins from the *MET3* promoter which is not subject to control by Pdr1.

## MET3-regulated Δ255–968 Pdr1 is a more effective transactivator than Δ255–968 D1082G Pdr1

Both the Δ255–968 and Δ255–968 D1082G *PDR1* genes were placed under control of the *MET3* promoter in a low-copy-number plasmid. The absence of Pdr1-dependent autoregulation in this construct was previously shown to rescue the lethality of the hyperactive Δ255–968 Pdr1 [10]. These two plasmids along with an empty vector control were introduced into *pdr1Δ* and *med15AΔ pdr1Δ* strains. Representative transformants were then assayed for the ability to

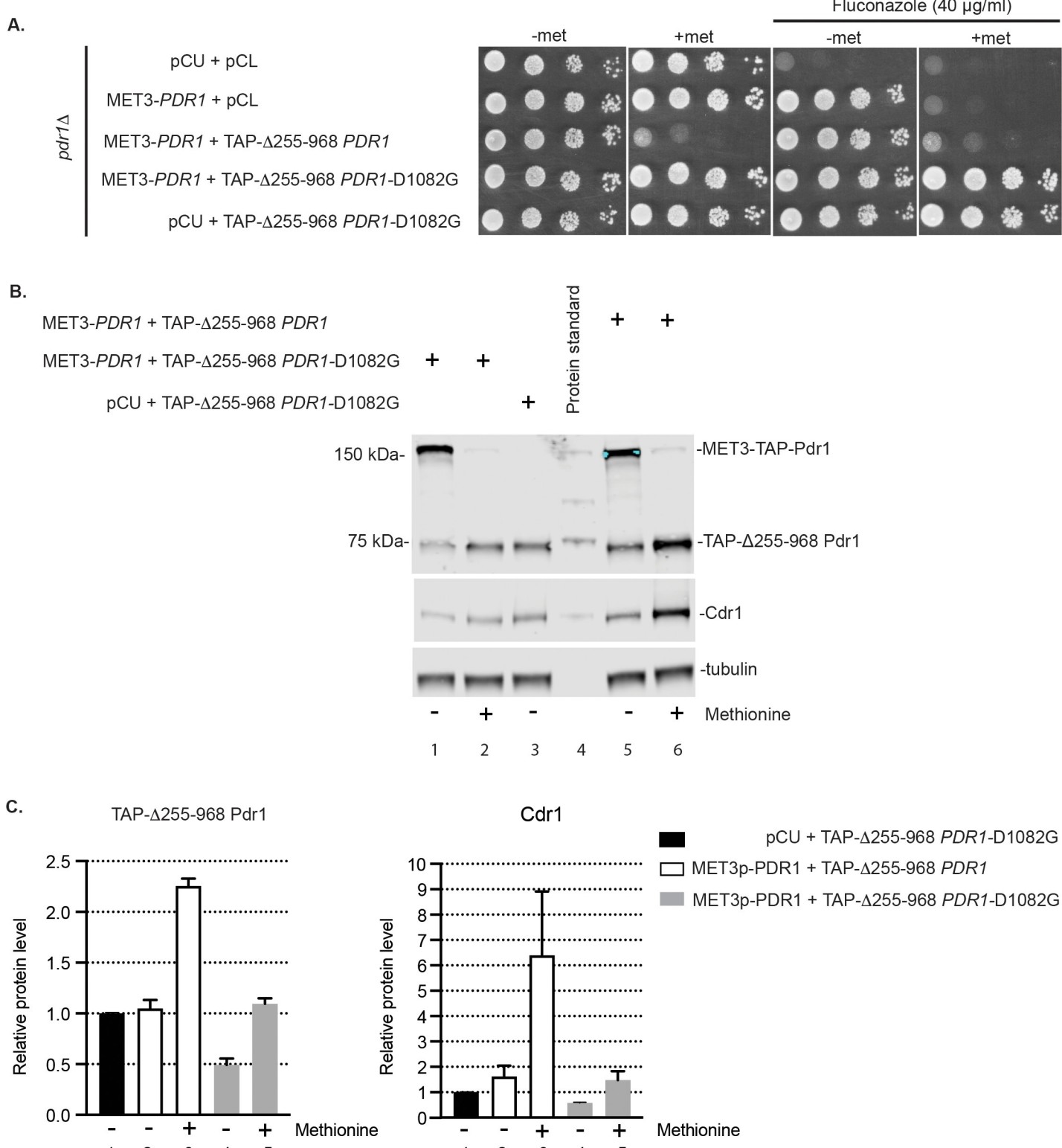

**Fig 5. Autoregulated mutant Δ255–968 Pdr1-D1082G drives less Cdr1 upon depletion of wild type Pdr1 than Δ255–968 Pdr1 form.** A. Strain *pdr1Δ* was co-transformed either with empty vectors pCU and pCL, *MET3*-driven TAP-*PDR1* and pCL, *MET3*-TAP-*PDR1* and TAP-Δ255–968 *PDR1* with or without D1082G or pCU and TAP-Δ255–968 *PDR1*-D1082G form. Cells were grown in liquid minimal media without methionine until mid-log phase and spotted on minimal media with (2 mM) or without methionine or containing 40 μg/ml of fluconazole to test cell viability and drug resistance. B. Cells co-expressing internal deletion Pdr1 form Δ255–

968 TAP-*PDR1* or TAP-Δ255–968 *PDR1*-D1082G and *MET3*-driven wild-type TAP-*PDR1* in the *pdr1Δ* background were pre-grown in minimal media without methionine and then inoculated into the same media with (2 mM) or without methionine and analyzed for levels of wild-type TAP-Pdr1, TAP-Δ255–968 Pdr1 and Cdr1 by western blotting using anti-Pdr1 or anti-Cdr1 antibodies. Tubulin was used as loading control. A representative western blot is shown. C. Relative TAP-Δ255–968 Pdr1 and Cdr1 protein level in the cells analyzed above. Levels of TAP-Pdr1 and Cdr1 in each strain were normalized to TAP-Pdr1 and Cdr1 levels in the strain co-expressing pCU vector and TAP-Δ255–968 Pdr1-D1082G. Error bars represent standard error of the mean.

drive fluconazole resistance as well as expression of the Δ255–968 Pdr1 derivatives and Cdr1 expression as above.

*MET3*-driven expression of either internally deleted form of Pdr1 led to high level fluconazole resistance in both genetic backgrounds (Fig 6A). However, the Δ255–968 D1082G Pdr1 derivative was less effective at supporting fluconazole resistance than the Δ255–968 Pdr1 in the *med15AΔ pdr1Δ* strain as evidenced by slight decrease in the growth when Δ255–968 D1082G Pdr1 is present.

To correlate this reduced drug resistance with an expression defect, we analyzed expression of the Δ255–968 Pdr1 forms of Pdr1 as well as Cdr1 (Fig 6B and 6C). Expression levels of the two different forms of Δ255–968 Pdr1 were equivalent when driven by the induced *MET3* promoter, although a slight reduction in level of the Δ255–968 D1082G Pdr1 form was detected. The positive role of the D1082 position could best be appreciated in the *med15AΔ* strain as in the presence of Med15A, transcription was already so high that the D1082 position was less important. A significant reduction in Cdr1 expression was seen when the Δ255–968 D1082G Pdr1 was present in the *med15AΔ pdr1Δ* strain compared to the Δ255–968 Pdr1 derivative. These results are consistent with the D1082G allele of Pdr1 reducing the ability of the C-terminal transactivation domain to elevate *CDR1* transcription to the same level as the Δ255–968 Pdr1 containing the normal D1082 residue.

## Complex interactions in the Pdr1 C-terminus support normal transactivation of gene expression

The unexpected reduction in transactivation function caused by a GOF form of Pdr1 led us to analyze in greater detail the structure of the C-terminal region of this transcription factor. We first carried out this analysis in the context of the Δ255–968 Pdr1 derivative to focus on the activity of the isolated transactivation domain. Previous work mapped an interaction interface between Med15A and Pdr1 to the C-terminus between residues 969 and 1107 [10]. Additionally, this same region contained a sequence called a 9 amino acid transactivation domain (9aa TAD) that has been argued to be universally recognized by the eukaryotic transcriptional machinery [23]. Mutagenesis experiments performed on the *Saccharomyces cerevisiae* Pdr1 (ScPdr1) that altered the cognate residues from 1097 to 1099 (numbering from CgPdr1) to alanine led to a loss of transcriptional activation function of this region [24]. We altered LWG (1097–1099) in Pdr1 to three alanine residues to test the role of this region in function of the *C. glabrata* protein. This triple substitution mutation is referred to as LWG1097AAA (abbreviated AAA) (Fig 7A). This AAA allele was analyzed separately but also in combination with the D1082G mutation. These mutants were introduced in low-copy-number plasmids into *pdr1Δ* or *med15AΔ pdr1Δ* strains and then tested for fluconazole resistance (Fig 7B) as well as expression of the Δ255–968 Pdr1 derivatives and Cdr1 by western blotting as described earlier (Fig 7C and 7D).

Upon transformation into the *pdr1Δ* strain, we recognized that the LWG1097AAA allele also rescued the lethality of the Δ255–968 Pdr1 as the D1082G was found to do earlier. The LWG1097AAA allele modestly lowered the level of drug resistance compared to D1082G in the absence of Med15A. Interestingly, when combined with the D1082G allele, this triple

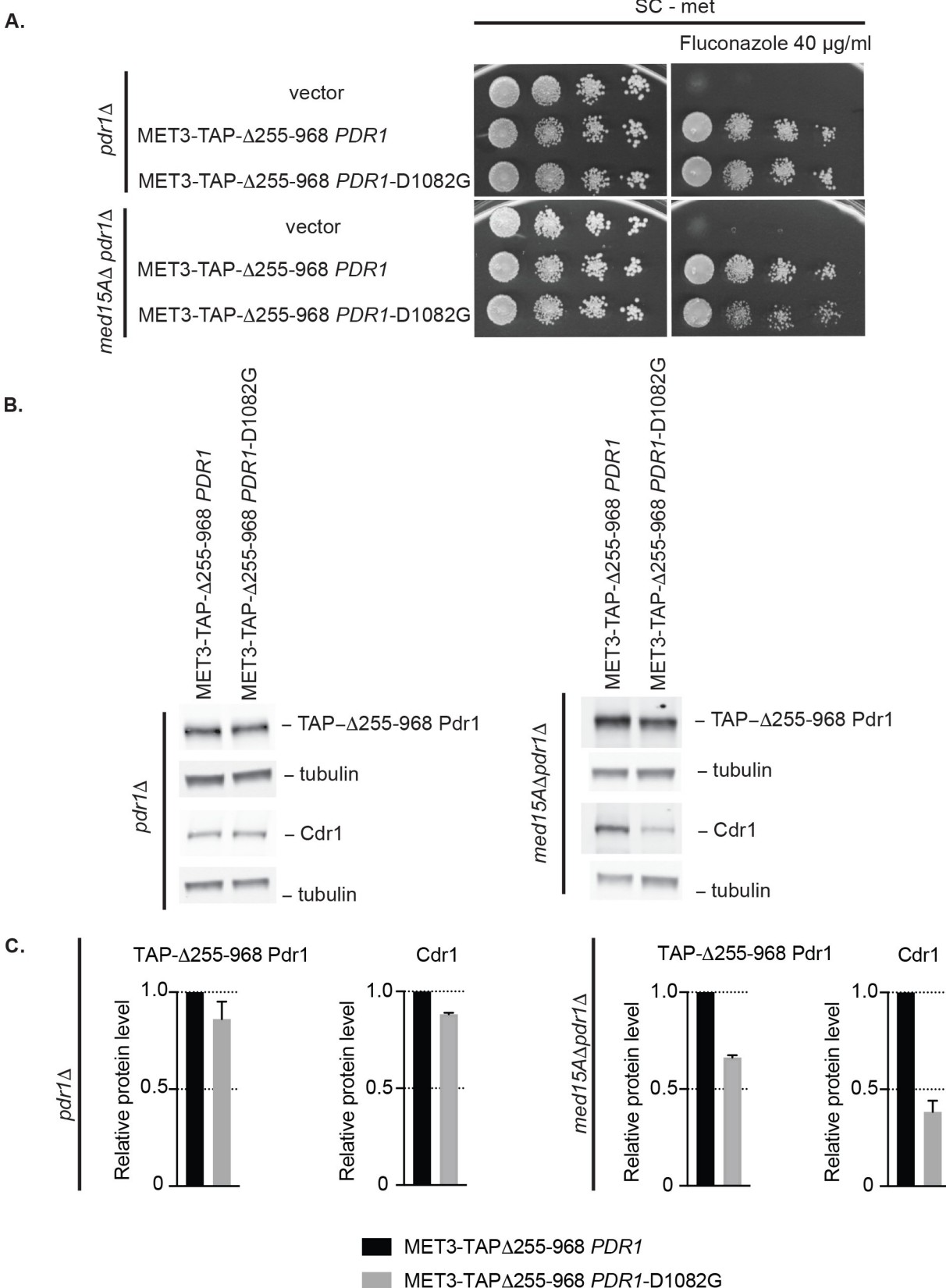

**Fig 6. *MET3*-driven mutant Δ255–968 Pdr1-D1082G induces 2-fold less Cdr1 than *MET3*-driven Δ255–968 Pdr1 in cells lacking the Med15A coactivator.** A. Strains *pdr1Δ* and *med15AΔ pdr1Δ* were transformed with vector only or the *MET3*-driven form of Δ255–968 *PDR1* with or without D1082G. Transformants were grown to mid-log phase in liquid minimal media without methionine and spotted on the same media containing or lacking 40 μg/ml of fluconazole. B. Mid-log cells expressing *MET3*-controlled forms of Δ255–968 *PDR1* with or without D1082G in *pdr1Δ* and med15AΔ *pdr1Δ* background were analyzed for levels of TAP-Δ255–968 Pdr1 and Cdr1 by western blotting using anti-Pdr1 or anti-Cdr1 antibodies. Tubulin was used as loading control. A representative western blot is shown. C. Relative TAP-Δ255–968 Pdr1 and Cdr1 protein level in the strains analyzed above. Protein levels in each strain were normalized to TAP-Δ255–968 Pdr1 and Cdr1 levels in the strain expressing *MET3*-Δ255–968 *PDR1*. Error bars represent standard error of the mean.

mutant showed a marked reduction in the level of fluconazole resistance developed. This effect of the triple mutant form of Δ255–968 Pdr1 was even more apparent in the *med15AΔ pdr1Δ* strain. In this genetic background, the presence of these two alterations in Pdr1, coupled with loss of the Med15A Mediator subunit, nearly eliminated Pdr1-driven fluconazole resistance.

Use of the *med15AΔ pdr1Δ* strain allowed the inclusion of the Δ255–968 Pdr1 as a comparator since loss of the Med15A subunit is known to suppress the lethality of this form of Pdr1 [10]. Each single mutant form tested here led to a decrease in fluconazole resistance, again supporting a role for these two sequence elements in transcriptional activation by Pdr1.

Analyses of expression of the Δ255–968 forms of Pdr1 and Cdr1 were consistent with the fluconazole resistance phenotypes described above. Loss of the Med15A Mediator subunit decreased expression of both Δ255–968 Pdr1 and Cdr1 in every case except the double mutant which remained at a low level whether Med15A was present or not. These data present a coherent picture of the D1082 and LWG1097 protein regions of Pdr1 acting as positive interfaces for transcriptional activation that are independent of the contribution of Med15A. We next moved these mutations back into of the full-length Pdr1 to analyze their effects in this context.

## Context-dependent effect of C-terminal mutations in Pdr1

The LWG1097AAA mutation was introduced into a wild-type copy of *PDR1* with or without the presence of the D1082G allele. These three mutant forms of Pdr1 were introduced on low-copy-number plasmids into either the *pdr1Δ* or *med15AΔ pdr1Δ* strains and tested for function as described above.

The LWG1097AAA mutant exhibited an increase in fluconazole tolerance compared to wild-type Pdr1 (Fig 8A). This was still significantly less drug resistant than the D1082G Pdr1 mutant. Fluconazole resistance of both of these single mutant forms of Pdr1 was highly Med15A-dependent as loss of this Mediator component caused a large increase in fluconazole susceptibility at higher drug concentrations. As seen above in the Δ255–968 Pdr1 derivatives, the presence of both the LWG1097AAA and D1082G mutations caused a striking increase in fluconazole susceptibility.

Expression of Cdr1 correlated well with the observed fluconazole resistance of the mutant strains (Fig 8B and 8C). The highest level of expression of both Pdr1 and Cdr1 was seen in the presence of the D1082G mutant as expected from previous work [10,11]. However, strong elevation of Cdr1 expression was found in the presence of the LWG1097AAA Pdr1, although this was markedly reduced by loss of Med15A. The double D1082G LWG1097AAA Pdr1 supported the lowest levels of Cdr1 expression among tested mutants, consistent with the low-level fluconazole resistance this protein supported.

Finally, we tested the ability of these different Pdr1 derivatives to recruit Med15A to the *CDR1* promoter using chromatin immunoprecipitation. A strain expressing an epitope-tagged form of Med15A was used to evaluate the level of this Mediator subunit associated with the *CDR1* promoter by standard techniques.

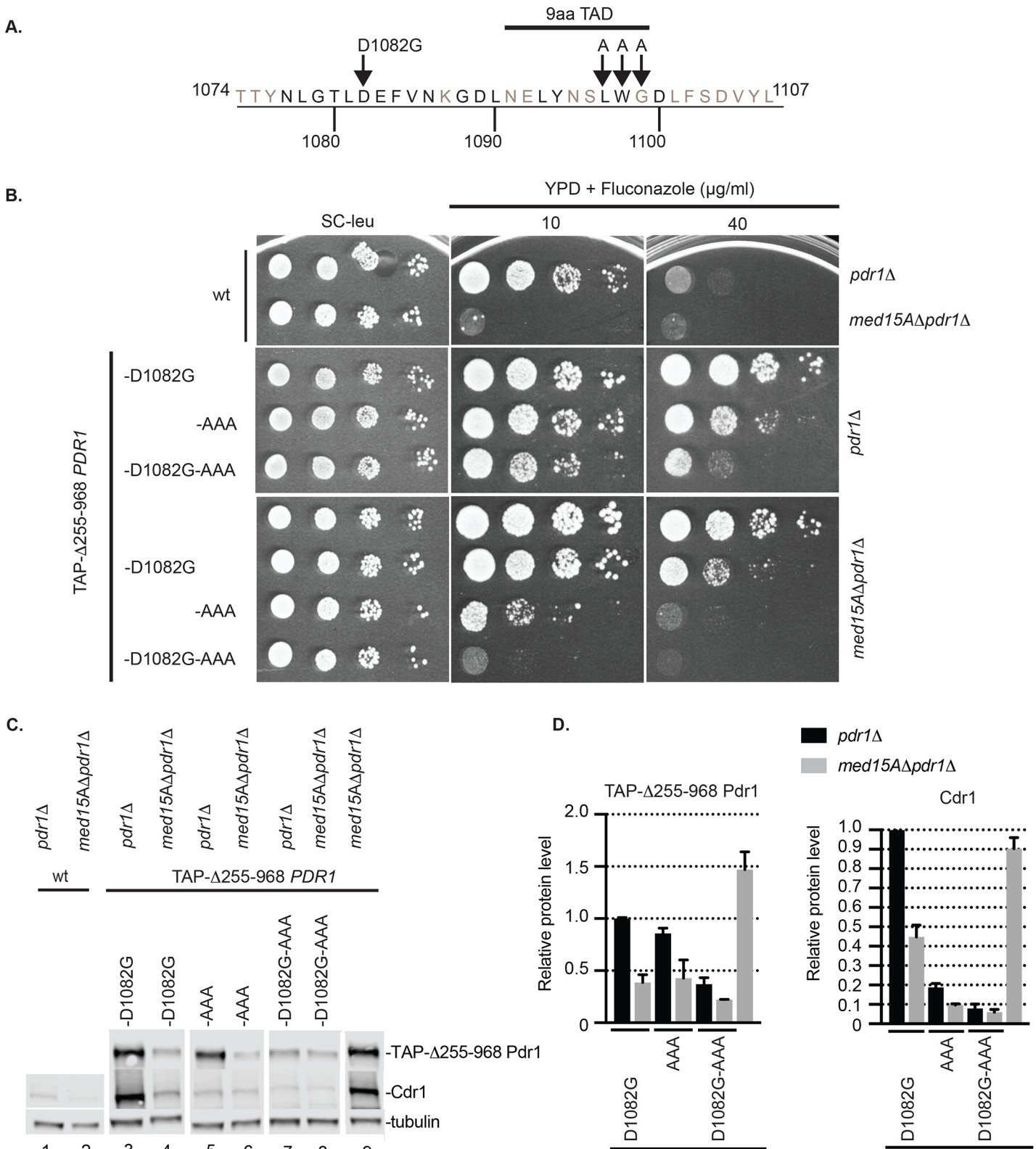

**Fig 7. D1082 region of transactivation domain of Pdr1 acts independently on 9aa TAD and Med15A to contribute to transactivation potential of Δ255–968 Pdr1.**
A. Amino acid sequence of the C-terminal end of Pdr1 transactivation domain. The position of D1082G mutation, 9aa transactivation domain (9aaTAD) and LWG residues within the 9aa TAD that were changed to alanines are shown. CgPdr1 amino acids conserved with *Saccharomyces cerevisiae* ScPdr1 and ScPdr3 are depicted in

black. B. Strain *pdr1Δ* and *med15AΔ pdr1Δ* were transformed with vectors carrying Δ255–968 *PDR1* with or without D1082G and AAA mutations. Transformants were grown to mid-log phase in liquid minimal selective media and spotted on the same media or YPD media containing 10 or 40 μg/ml of fluconazole to test for drug resistance. C. Mid-log cells expressing Δ255–968 *PDR1* mutant forms from Panel A were analyzed for levels of TAP-Δ255–968 Pdr1 and Cdr1 by western blotting using anti-Pdr1 or anti-Cdr1 antibodies. Tubulin was used as loading control. A representative western blot is shown. D. Relative Δ255–968 Pdr1 and Cdr1 protein level in the strains analyzed above. Protein levels in each strain were normalized to levels from the strain expressing Δ255–968 Pdr1-D1082G in the *pdr1Δ* background. Error bars represent standard error of the mean.

The presence of the D1082G Pdr1 polypeptide supported the highest level of *CDR1* promoter that could be recovered in the anti-Med15A ChIP reaction (Fig 8D). The presence of the LWG1097AAA Pdr1 exhibited a modest increase in Med15A association with *CDR1*. Importantly, the double D1082G LWG1097AAA Pdr1 was reduced to a similar level to the LWG1097AAA Pdr1 single mutant, indicating that the enhanced Med15A association seen in the D1082G Pdr1 mutant depends on the presence of the wild-type sequences at the LWG1097 region.

## Discussion

Experiments from several laboratories have provided strong evidence that the transcriptional activation function of Pdr1 is elevated in fluconazole resistant mutants isolated from the clinic [9,11,16–18]. Since *PDR1* is an autoregulated gene, it is difficult to ensure that the enhanced transcription of Pdr1-regulated genes is not solely due to the linked increase in expression of *PDR1* itself seen in these GOF mutant strains rather than increased activity of each polypeptide chain. Here we demonstrate that expression of two different GOF forms of Pdr1 from a promoter lacking autoregulation still leads to increased expression of target genes. These data support the view that GOF forms of Pdr1 accumulate to higher levels in the cell and also possess increased transcriptional activation function than the wild-type factor.

When driven from the *MET3* promoter, GOF forms accumulated to a lower level than the wild-type protein yet target gene transcription was higher. Our previous data have indicated that the Bre5 protein appears to be a negative regulator of Pdr1 levels in *C. glabrata* and strains lacking Bre5 accumulated higher levels of Pdr1 than wild-type cells [15]. Additionally, GOF forms of Pdr1 were degraded at a faster rate than the wild-type protein [10]. These data suggest that an important step in control of Pdr1 activity is regulation of the proteolysis of this factor. Even with the greater rate of turnover of these GOF proteins, their intrinsic capability to induce gene expression is greater than the normal factor.

The enhanced transactivation capacity of GOF forms of Pdr1 could most simply be explained by each of these different mutations inactivating some common negatively acting domain that would otherwise restrain Pdr1 target gene induction. However, the genetic analyses of different GOF alleles argues against this simple interpretation. When the three CRD GOF mutations were combined into the Triple allele, the resulting mutant protein produced the highest level of fluconazole resistance and Cdr1 expression in the absence of Med15A by comparison to any single mutant form we have analyzed. Strikingly, combination of D1082G with the Triple mutant (Quadruple Pdr1) exhibited less transactivation capability than the isogenic Triple. We were concerned that generation of the Quadruple mutant would have a lethal phenotype as was seen for the Δ255–968 Pdr1 [10] but this was not the case. This unexpected result prompted us to analyze the role of the D1082 region in greater detail by testing its function in the context of the Δ255–968 Pdr1 derivative.

Based on our previous analyses of the Δ255–968 Pdr1 protein [10], we believe this mutant factor consists of the DNA-binding domain linked to the transcriptional activation domain with negative regulatory information removed. The finding that the D1082G mutation rescued the lethality of this mutant Pdr1 protein was unexpected and argued that, like other rescuing

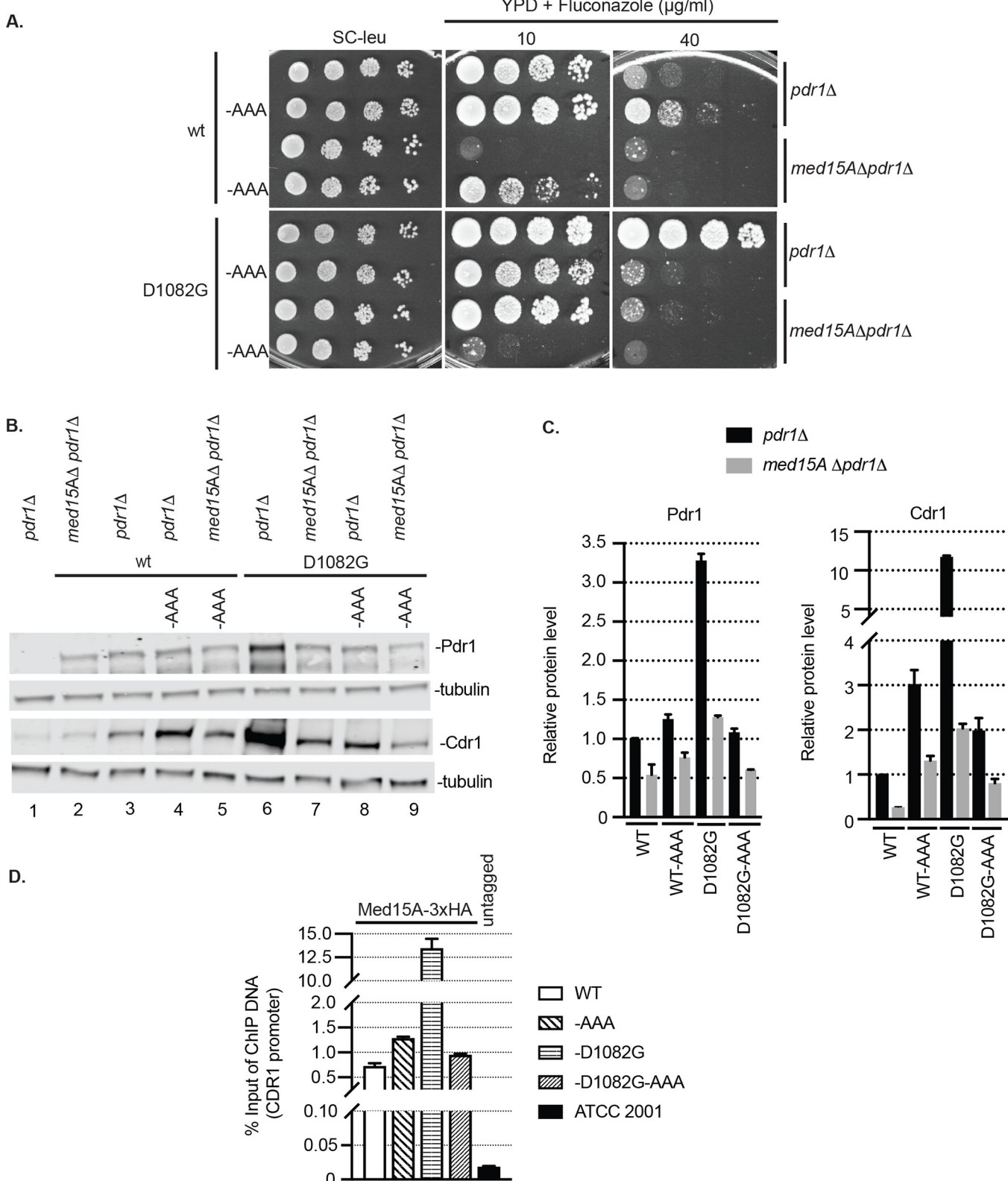

**Fig 8. Transactivation via the TAD of full-length and Δ255–968 Pdr1 depend on similar network of coactivators.** A. Strain *pdr1Δ* and *med15AΔ pdr1Δ* were transformed with vectors carrying full length *PDR1* forms with or without D1082G and AAA mutations. Transformants were grown to mid-log phase in liquid minimal selective media and spotted on the same media or YPD media containing 10 or 40 μg/ml of fluconazole to test for drug resistance. B. Mid-log

cells expressing full length Pdr1 forms from Panel A were subjected to protein extraction and analyzed for levels of Pdr1 and Cdr1 by western blotting using anti-Pdr1 or anti-Cdr1 antibodies. Tubulin was used as loading control. A representative western blot is shown. C. Relative Pdr1 and Cdr1 protein level in the strains analyzed above. Protein levels in each strain were normalized to levels from the strain expressing wild type Pdr1 in *pdr1Δ* background. Error bars represent standard error of the mean. D. Protein-DNA complexes in mid-log *pdr1Δ* cells expressing wild-type, D1082G, AAA or D1082G-AAA mutant *PDR1* versions were crosslinked with formaldehyde and the chromatin from the cell lysate was sheared. Chromatin immunoprecipitation was used to purify Med15A-3xHA with associated DNA using anti-HA antibody. The purified DNA was analyzed by qPCR for the presence and abundance of *CDR1* promoter using the set of primers specific to this region. A strain carrying the untagged version of Med15A (ATCC 2001) was used as control for anti-HA antibody specificity. The percentage of input method was used to quantify the amount of *CDR1* promoter pulled-down by Med15A-3X HA. Measurements represent the result of two independent experiments performed on two sets of transformants and the error bars were calculated as standard error of the mean.

alleles, D1082G reduced the function of this hyperactive transcription factor [10] and as a consequence has been well tolerated in *pdr1Δ* cells. This was in contrast to the effect of D1082G on full-length Pdr1 which caused a strong GOF phenotype that led to its initial isolation [11]. These context-dependent effects were the first indication that the effect of D1082G on Pdr1 was not as straightforward as originally thought. We found similar complex behavior for a mutation in the 9aa TAD region in Pdr1. Based on previous experiments on ScPdr1 [24], we anticipated that the LWG1097AAA mutant would exhibit reduced transcriptional activation. This result was observed when the Pdr1 transactivation domain was assayed in the context of Δ255–968 Pdr1 but surprisingly, this same LWG1097AAA mutant also behaved as a GOF allele in the full-length protein. In these two manners, the two different C-terminal mutants behaved very similarly.

Together, these findings support the hypothesis that the carboxy-terminal mutants define a complex region of Pdr1 that has both positively and negatively acting functions in terms of transcriptional regulation. A simple model for the function of the D1082 position would be that its conversion to a glycine converts the resulting mutant protein into a high activity state transcription factor. Our data indicate that this is not the only effect of the D1082G lesion as it also reduces the ability of the isolated transcriptional activation domain to stimulate gene expression as measured in the Δ255–968 Pdr1 protein. This same type of bifunctional role was seen for the LWG1097AAA mutant. It is important to note that there are a wide range of amino acid changes that span the entire C-terminal region that lead to Pdr1 being locked in a hyperactive state [11]. We speculate that extensive interactions are required to keep Pdr1 in its low activity conformation. Interruption of these interactions by a broad range of different amino acid substitutions cause the protein to exhibit a constitutive high-level target gene transcriptional phenotype.

Since the D1082 and LWG1097 regions of the Pdr1 C-terminus affect transcription of Pdr1 in both a positive and negative manner, we propose the model shown in Fig 9 as a working hypothesis for regulation of this factor. The C-terminal activation domain of Pdr1 (residues 968–1107) may be released from intramolecular repression by changes such as D1082G, LWG1097AAA, multiple mutations in the CRD or deletion of the entire CRD as in the internal deletion Δ255–968 Pdr1. It is also possible that interactions with the trans-acting negative regulators Bre5 and Jjj1 might also be involved in the effects seen for the impact of C-terminal mutagenesis [14,15]. We speculate that the D1082 and LWG1097 regions also have positive interactions with co-activators other than Med15A as the combination of mutational alterations in these two regions, coupled with deletion of *MED15A*, caused the most profound defect in Pdr1-mediated fluconazole resistance and Cdr1 expression (Fig 8). These observations lead us to propose a bifunctional character to these two regions in the Pdr1 C-terminus.

It is also important to compare the effects of the Triple mutations in the CRD with the deletion of this region. Based on the fact that the complete deletion of the CRD from Pdr1 produced a lethal transcriptional activator, we predicted that point mutants that eliminated all negative regulation of the CRD would have a similar phenotype. Even when 3 different GOF

mutants were combined to form the Triple mutant, this form of Pdr1 could be maintained by the cell. This finding argues that additional negatively acting signals must still be present in the Triple mutant Pdr1 that allow sufficient repression of Pdr1 activity to be maintained to avoid the lethal phenotype seen for Δ255–968 Pdr1. Identification of the negative inputs modulating Pdr1 activity are a high priority of current research.

Finally, comparison of the C-terminus of Pdr1 with the transcriptional activation domain of Gcn4 [25] suggests conservation of a series of hydrophobic residues as shown in Fig 9B. Our mutation of the putative 9aa TAD changed two of these conserved residues to alanine and caused a defect in transactivation of the isolated C-terminus in the Δ255–968 Pdr1 context (Fig 7B). This was also seen when the LWG1097AAA allele was combined with D1082G in the full-length Pdr1. This compound mutation exhibited a large decrease in expression of Cdr1, fluconazole resistance and the ability to recruit Med15A to the *CDR1* promoter. These data are most consistent with the C-terminal Pdr1 TAD engaging in multivalent interactions with the transcriptional machinery. Loss of any one of these interactions is not sufficient to completely block Pdr1 activation of gene expression. This was also shown by the discovery of a small compound inhibitor of Pdr1 gene activation [26] which was not equally effective at lowering the azole resistance of all GOF alleles of *PDR1*. Evidence was presented that this inhibitor did interfere with Med15A gene activation. These data are consistent with our findings that Med15A is an important but not exclusive co-activator for Pdr1. Identification of these additional co-activator interactions is an important step towards understanding the complex mechanisms underlying gene activation by this central mediator of azole resistance in *C. glabrata* and generating new therapeutic targets.

## Materials and methods

### Strains and growth conditions

All strains used in the study are listed in Table 1. Cultures of *C. glabrata* were grown at 30˚C. Cells grown overnight for 16 hours were diluted to $OD_{600}$ = 0.2 and grown to mid-log phase of $OD_{600}$ = ~1 for all experiments. Complete YPD (yeast extract 1%, peptone 2%, glucose 2%) medium was used for non-selective growth and drug treatments. Minimal SD media (yeast nitrogen base with ammonium sulfate) supplemented with appropriate amino acids were used for selective growth of *C. glabrata* strains transformed with plasmids. Methionine at a final concentration of 2 mM was added to minimal SD media in experiments using the *MET3* promoter to repress expression of the downstream gene.

### Plasmids

All plasmids used in this study are listed in Table 2. Cloning into plasmids was done using Gibson Assembly cloning kit (NEB #E5510S) and the identity of recombinant vectors was verified with restriction enzyme analysis of plasmid DNA and sequencing. *PDR1* alleles were cloned into the low-copy-number vectors originating from plasmids pCU (*URA3*) or pSK60 (*LEU2*) that contain *C. glabrata* CEN/ARS replication origin [19]. To generate *MET3*-driven wild type *PDR1*, R376W and D1082G alleles, the *MET3* promoter was amplified from the pCU vector and cloned upstream of the *PDR1* coding sequence in the pSK61 vector (pLS6), D1082G *PDR1* in pSK70 vector (pLS7) or R376W *PDR1* in pSK71 vector (pLS8). To generate R376W-Y584C-P822L-D1082G *PDR1* allele (Quadruple), the DNA region containing the R376W mutation was amplified from pSK71, the region with Y584C was amplified from pSK74 and the region with P822L was amplified from pSK68. All three PCR fragments were cloned into the pSK70 plasmid containing D1082G mutation (pLS10). To generate the R376W/Y584C/P822L *PDR1* allele (Triple), the DNA region of Quadruple *PDR1* allele containing the

**A.**

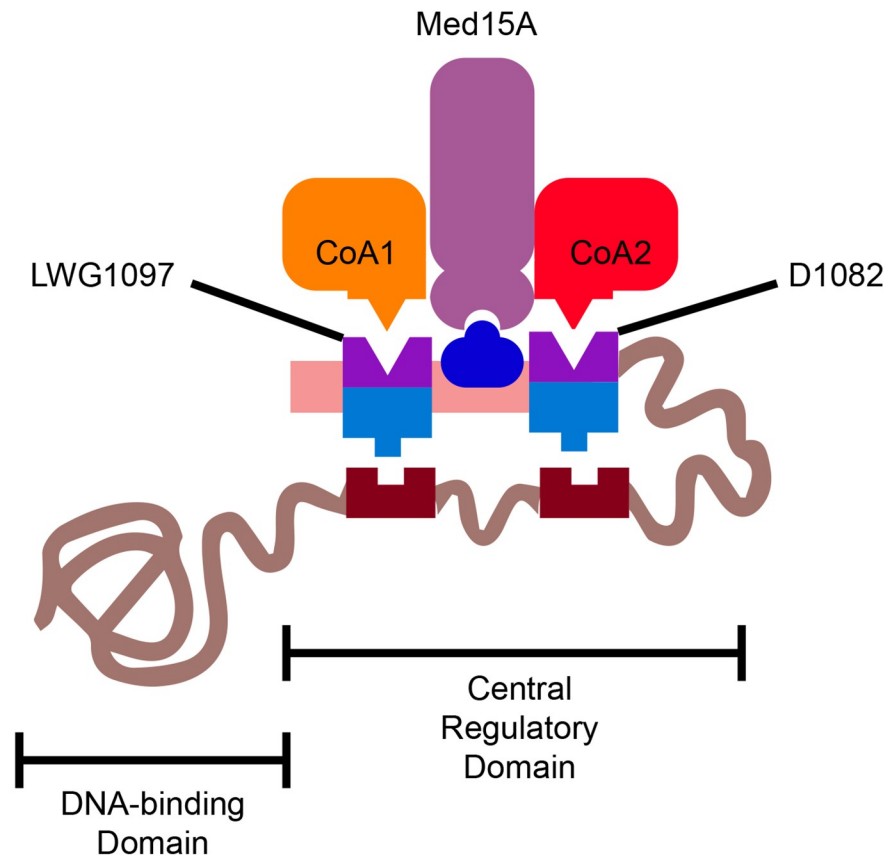

**B.**

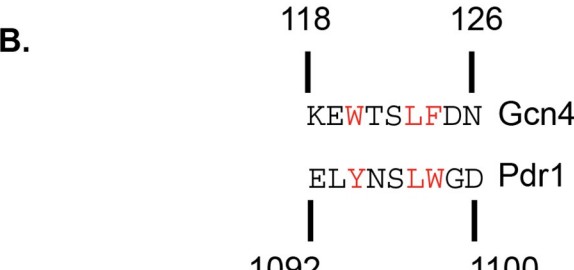

**Fig 9. Regulatory model for control of Pdr1 transactivation.** A. A hypothetical model for the regulation of the Pdr1 C-terminal TAD is shown. Location of the N-terminal DNA-binding domain and the central regulatory domains are indicated. The positive interactions of the C-terminal TAD are indicated by the interactions with co-activator proteins (CoA1 and CoA2) as well as Med15A. Negative regulatory interactions are shown as the interactions with the CRD by the same regions. This model proposes that inhibition of TAD function is accomplished via interactions between the

C-terminal TAD and the CRD but binding of trans-acting protein factors remains a possibility. B. Conservation of hydrophobic residues between a region of the Gcn4 transactivator and the Pdr1 TAD. The residues in red are crucial for activation via ScGcn4 [25] and a similar region in Pdr1 is indicated in red.

mutations R376W/Y584C/P822L was amplified from pLS10 and cloned into the pSK61 plasmid (pLS9) containing the wild-type D1082 site. The internal deletion derivative allele Δ255–968 *PDR1* containing the D1082G mutation was constructed by amplifying the DNA region containing the D1082G mutation from the plasmid pSK70. The corresponding PCR fragment was cloned into the *URA3*-marked plasmid pLS2 where it replaced the wild-type form of the *PDR1* transactivation domain (pLS11). The same strategy was used to generate internal deletion Δ255–968 *PDR1* containing the D1082G mutation in the *LEU2*-marked plasmid pLS3. The *MET3*-driven form of Δ255–968 *PDR1* with the D1082G mutation was prepared by PCR amplifying the TAP-Δ255–968 *PDR1*-D1082G part from the vector pLS11 and the PCR product was cloned into pCU plasmid under the control of *MET3* promoter (pLS13). To make a plasmid that carries the TAP-Δ255–968 *PDR1* containing the LWG1097AAA mutation (AAA mutant) in the 9aa transactivation domain of *PDR1*, the mutations were first introduced into the sequence of TAP-Δ255–968 *PDR1* in pLS2 plasmid by site-directed PCR mutagenesis. Introduction of the AAA mutation was confirmed by DNA sequencing. The *PDR1* region containing the AAA mutation was then amplified and subcloned into the original pLS2 to yield pLS14. Introduction of the LWG1097AAA mutation generates a new AfeI restriction site that was confirmed by restriction analysis of pLS14. TAP-Δ255-968-D1082G-AAA *PDR1* was prepared in a similar manner and the PCR fragment from the *PDR1* region containing the D1082G and AAA mutations was cloned into the plasmid pLS11. To generate the LWG1097AAA mutation in the wild-type *PDR1* or D1082G-*PDR1* allele, the mutations were first introduced into the sequence of pSK61 (WT) or pSK70 (D1082G) by site-directed PCR mutagenesis as described above. After verification of the presence of the LWG1097AAA allele, the sequence of *PDR1* around the AAA region was PCR-amplified and subcloned into pSK61 (to generate pLS16) or pSK70 (to generate pLS17). To make pLS18, the coding sequence of *MED15A* with 800 bp of native promoter was amplified from the genome of SPG96 and cloned into the pSK60 vector along with a PCR fragment containing three copies of human influenza hemagglutinin tag (3xHA) that was amplified from the pFA6a plasmid series [27] and was used to generate a C-terminal fusion gene *MED15A-3xHA*. This fusion gene with 800 bp of *MED15A* promoter was then subcloned into the pCU plasmid to make pLS18.

## Transformation of *C. glabrata*

The yeast transformation was performed using the lithium acetate method [28]. 3 OD$_{600}$ units of mid-log cells and five hundred nanograms of plasmid DNA were used per transformation reaction. Cells were exposed to heat shock at 42°C for 1 hour, plated on selective minimal SD media and incubated at 30°C for 2 days.

**Table 1. List of strains.**

| Name | Background | Genotype | Reference |
|------|-----------|----------|-----------|
| MRY822 | ATCC 2001 | *pdr1Δ::natMX4 his3Δ::*FRT *leu2Δ::*FRT *trp1Δ::*FRT | [20] |
| SPG96 | ATCC 2001 | *his3Δ::*FRT *leu2Δ::*FRT *trp1Δ::*FRT *ura3Δ(-85+932)::*Tn*903NeoR* | [10] |
| SKY107 | ATCC 2001 | *pdr1Δ::natMX4 his3Δ::*FRT *leu2Δ::*FRT *trp1Δ::*FRT *ura3Δ(-85+932)::*Tn*903NeoR* | [10] |
| LSY2 | ATCC 2001 | *pdr1Δ::natMX4 his3Δ::*FRT *leu2Δ::*FRT *trp1Δ::*FRT *ura3Δ(-85+932)::*Tn*903NeoR med15AΔ::HIS3MX6* | [10] |

## Drug treatment

Mid-log cells were spotted in ten-fold serial dilutions on YPD or minimal SD media containing the indicated concentrations of fluconazole (LKT Laboratories, Inc. #F4682).

## Real time qPCR

5 $OD_{600}$ units of mid-log cells were used per sample. Total RNA was extracted using Trizol reagent (Invitrogen #15596026) and chloroform. RNA was then purified with RNeasy minicolumns (Qiagen #74104) and 500 nanograms were reverse-transcribed using iScript cDNA synthesis kit (Bio-Rad, #1708890). qPCR was performed using iTaq universal SYBR green supermix (Bio-Rad #1725121). The average Ct value for each sample was calculated from the triplicate. Cg*TEF1* gene was used for normalization of variable cDNA levels. Wild type Pdr1 values were used to normalize the drug-efflux pump levels among strains. A comparative $2^{-\Delta\Delta Ct}$ method was used to calculate the fold change of the gene of interest between samples [29]. All measurements represent the result of two independent experiments performed on two sets of transformants and the error bars were calculated as standard error of the mean.

## Western blot analysis

3 $OD_{600}$ units of mid-log cells were used per sample. Proteins were extracted as previously described [30], resuspended in urea sample buffer (8 M urea, 1% 2-mercaptoethanol, 40 mM Tris-HCl pH 8.0, 5% SDS, bromophenol blue) and incubated at 37˚C for 1 hour. The resuspended proteins were boiled at 90˚C for 10 minutes if Pdr1 levels were being analyzed and aliquots were resolved on precast ExpressPlus 4–12% gradient gel (GenScript #M41212). Proteins were electroblotted to nitrocellulose membrane, blocked with 5% nonfat dry milk and probed with anti-Pdr1 antibody [20] or anti-Cdr1 antibody [31]. All membranes were also probed for tubulin with 12G10 anti-alpha-tubulin monoclonal antibody (Developmental Studies Hybridoma Bank at the University of Iowa). Secondary Li-Cor antibodies IRD dye 680RD goat anti-rabbit (#926–68021) and IRD dye 800LT goat anti-mouse (#926–32210) were used in combination with the Li-Cor infrared imaging system (application software version 3.0) and Image Studio Lite software (Li-Cor) to detect and quantify the signal from the western blot. The relative protein levels of Pdr1 or Cdr1 were normalized to tubulin levels of the corresponding strain and then compared to reference strain. All measurements represent the result of two independent experiments performed on two sets of transformants and the error bars were calculated as standard error of the mean.

## Chromatin immunoprecipitation and qRT-PCR

The chromatin immunoprecipitation (ChIP) experiment was performed as previously described [31]. 50 $OD_{600}$ units of mid-log cells were treated with 1% formaldehyde to crosslink the DNA to proteins and the reaction was inhibited by adding glycine. Cells were washed with PBS and resuspended in FA-lysis buffer (50 mM HEPES-KOH, 140 mM NaCl, 1 mM EDTA, 1% Triton X-100, 0.1% sodium deoxycholate, supplemented with 1 mM PMSF and 1x Complete protease inhibitor). Glass beads (0.5 mm) were added to the cell suspension and the sample was vortexed at 4˚C for 2 minutes, five times with a 1 minute pause on ice between cycles. The beads were separated from the remaining sample and the cell lysate was split into AFA Fiber Pre-Slit Snap-Cap (6x15 mm) microtubes (Covaris) (130 μl sample per tube). Chromatin was sheared with E220 Focused-ultrasonicator (Covaris) under the following conditions: peak incident power (W): 175, duty factor: 20%, cycles per burst: 200, treatment time (sec): 720, temperature (˚C): 7, sample volume (μl): 130, under the presence of E220-intensifier

**Table 2. List of plasmids.**

| Name | Relevant features | Reference |
|------|-------------------|-----------|
| pSK60 | Sc*LEU2* Cg CEN/ARS | [10] |
| pCU | Sc*URA3* CgCEN/ARS | [19] |
| pSP76 | *PDR1* in pCU | [10] |
| pLS1 | *TAP-PDR1* in pCU | [10] |
| pSK61 | *PDR1* in pSK60 | [10] |
| pSK68 | P822L *PDR1* in pSK60 | [10] |
| pSK70 | D1082G *PDR1* in pSK60 | [10] |
| pSK71 | R376W *PDR1* in pSK60 | [10] |
| pSK74 | Y584C *PDR1* in pSK60 | [10] |
| pLS6 | *MET3-PDR1* in pSK60 | This study |
| pLS7 | *MET3*-D1082G *PDR1* in pSK60 | This study |
| pLS8 | *MET3*-R376W *PDR1* in pSK60 | This study |
| pLS9 | R376W-Y584C-P822L *PDR1* (Triple) in pSK60 | This study |
| pLS10 | R376W-Y584C-P822L-D1082G *PDR1* (Quadruple) in pSK60 | This study |
| pLS2 | TAP-Δ255–968 *PDR1* in pCU | [10] |
| pLS3 | TAP-Δ255–968 *PDR1* in pSK60 | [10] |
| pLS11 | TAP-Δ255-968-D1082G *PDR1* in pCU | This study |
| pLS12 | TAP-Δ255-968-D1082G *PDR1* in pSK60 | This study |
| pLS4 | *MET3*-TAP-*PDR1* in pCU | [10] |
| pLS5 | *MET3*-TAP-Δ255–968 *PDR1* in pCU | [10] |
| pLS13 | *MET3*-TAP-Δ255-968-D1082G *PDR1* in pCU | This study |
| pLS14 | TAP-Δ255-968-AAA *PDR1* in pCU | This study |
| pLS15 | TAP-Δ255-968-D1082G-AAA *PDR1* in pCU | This study |
| pLS16 | AAA *PDR1* in pSK60 | This study |
| pLS17 | D1082G-AAA *PDR1* in pSK60 | This study |
| pLS18 | *MED15A-3xHA* in pCU | This study |

(pn500141). Next, the supernatant (lysate) was separated from the rest of the sample and a fraction was removed as input control for chromatin immunoprecipitation and qRT-PCR. For immunoprecipitation, anti-HA monoclonal antibody (2–2.214) (Invitrogen) was used (1:100) against Med15-3xHA tagged protein. The antibody was first preincubated with the lysate for 2 hours at 4˚C. Next, the antibody-lysate combination was added to Dynabeads Protein G magnetic beads (Invitrogen) and the sample was incubated overnight at 4˚C on a nutator. Washing and all subsequent steps were performed as described previously [32]. To perform qRT-PCR on the purified ChIP-ed DNA, the primer pair specific for the region of *CDR1* promoter (-322 to -613 relative to start codon) was used. The strain ATCC 2001, that carries only an untagged, wild-type copy of Med15A was used as a negative control for the experiment. Each sample was analyzed in triplicate. 0.5 μl of ChIP-ed DNA or 20-fold diluted input DNA was used in the reaction with the final volume of 20 μl. Primers were used at a final concentration of 0.4 μM and the SYBR green master mix (Bio-Rad) was used as recommended by the manufacturer. The PCR reaction was carried out under the following conditions: One cycle of 95˚C for 30 seconds, 40 cycles of 95˚C for 15 seconds and 56˚C for 30 seconds on MyiQ2 BioRad device. To calculate the signal of enrichment of DNA from the *CDR1* promoter, the percent input method was applied. The data represent the result of two independent experiments performed on two sets of transformants and the error bars were calculated as standard error of the mean.

## Supporting information

**S1 Table. Quantitative Data used to generate figures.** All values used to generate data shown in Figures described here are provided in this table.
(XLSX)

## Acknowledgments

We thank Drs. Bao Vu, Sanjoy Paul and Damian Krysan for helpful discussions during this work.

## Author Contributions

**Conceptualization:** Lucia Simonicova, W. Scott Moye-Rowley.

**Formal analysis:** W. Scott Moye-Rowley.

**Funding acquisition:** W. Scott Moye-Rowley.

**Investigation:** Lucia Simonicova.

**Methodology:** Lucia Simonicova.

**Project administration:** W. Scott Moye-Rowley.

**Supervision:** W. Scott Moye-Rowley.

**Writing – original draft:** Lucia Simonicova.

**Writing – review & editing:** Lucia Simonicova, W. Scott Moye-Rowley.

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
