## [Decision Letter · Decision Letter 0]

9 Apr 2020

Dear Dr Moye-Rowley,

Thank you very much for submitting your Research Article entitled 'Functional information from clinically-derived drug resistant forms of the Candida glabrata Pdr1 transcription factor' to PLOS Genetics. Your manuscript was fully evaluated at the editorial level and by independent peer reviewers. The reviewers appreciated the attention to an important problem, but raised some substantial concerns about the current manuscript. Based on the reviews, we will not be able to accept this version of the manuscript, but we would be willing to review again a much-revised version. We cannot, of course, promise publication at that time.

You will see that the reviewers asked for clarifying editing or rewriting, and also for rethinking some of the main conclusions.  The suggestion (marked with ***) to look at a couple of additional target genes - CDR2 and SNQ2 - makes good sense as well.

If you decide to revise the manuscript for further consideration at PLOS Genetics, please aim to resubmit within the next 60 days, unless it will take extra time to address the concerns of the reviewers, in which case we would appreciate an expected resubmission date by email to plosgenetics@plos.org.

[LINK]

We are sorry that we cannot be more positive about your manuscript at this stage. Please do not hesitate to contact us if you have any concerns or questions.

Yours sincerely,

Aaron P. Mitchell, PhD

Guest Editor

PLOS Genetics

Gregory P. Copenhaver

Editor-in-Chief

PLOS Genetics

Reviewer's Responses to Questions

**Comments to the Authors:**

Reviewer #1: Gain-of-function (GOF) mutations (usually caused by single nucleotide substitutions that lead to amino acid exchanges) in the transcription factor Pdr1 result in constitutive overexpression of genes encoding multidrug efflux proteins and are the major mechanism of azole resistance in the pathogenic yeast Candida glabrata. In the absence of inducing signals, wild-type Pdr1 is kept in an inactive state by a central regulatory domain (CRD) that prevents the interaction of the C-terminally located activation domain (AD) with the transcriptional apparatus. GOF mutations that relieve Pdr1 from repression are located in the CRD or in the AD. Deletion of the CRD also results in a hyperactive Pdr1, which is toxic to the cells when expressed from its own promoter because of the high Pdr1 levels achieved by autoregulation. In addition to the CRD, trans-acting negative regulators are also involved in controlling Pdr1 activity. In the present study, the authors aimed to elucidate how different GOF mutations in Pdr1 activate the transcription factor. For this purpose, they compared the effect of different GOF mutations and combinations thereof on Pdr1 activity and dependence on the mediator subunit Med15A. They found that mutations in the AD that cause constitutive activation of Pdr1 reduce the activity of the transcription factor when combined with mutations in the CRD and conclude that the AD contains different regions that interact with trans-regulatory factors and have both positive and negative regulatory functions.

This thoroughly performed, detailed study provides interesting information about the effects of different Pdr1 mutations on the activity of the transcription factor. However, I do not fully agree with some of the authors’ conclusions, and they may consider the following comments.

General comments

1) The authors found that the strongly activating D1082G GOF mutation, as well as the unexpectedly activating LWG1097AAA mutation, in the AD actually reduce the activity of hyperactive forms of Pdr1 that already contain other activating mutations (GOF mutations in or deletion of the CRD). From this observation they conclude that the C-terminal AD has both positive and negative regulatory functions. In my opinion, there is a much simpler explanation for the findings. It is reasonable to assume that the D1082G and LWG1097AAA mutations have a negative effect on the functionality of the AD but at the same time prevent its interaction with the CRD, resulting in a derepressed state. The negative effect of the AD mutations is only seen when Pdr1 is already activated by other alterations that block interaction of the AD with the CRD. This is analogous to, for example, resistance mutations in a drug target enzyme, such as Erg11. Here, mutations that confer resistance to an inhibitor (in this case azole drugs) often reduce the enzymatic activity in the absence of the inhibitor but allow sufficient residual activity in its presence. One would not argue that the corresponding region in the wild-type enzyme negatively affects enzyme activity.

2) The authors argue that increased Pdr1 function upon the cumulative addition of different GOF mutations indicates that the mutations affect different regulatory mechanisms (see lines 188-193). While this may be the case, it is also possible that each mutation only partially reduces the same repression mechanism and they have an additive effect. Even the triple mutation in the CRD might still allow some interaction with the AD, explaining why it is not as toxic as the CRD deletion.

3) Does the fact that both the D1082G mutation and the LWG1097AAA mutation activate wild-type Pdr1 and reduce the activity of Pdr1 containing other GOF mutations prove that they define two separate regions interacting with the CRD as well as with different coactivators (see also the model in Fig. 9A)? Is it not possible that both mutations result in conformational changes that alter the interaction of the AD (at whatever sites) with the CRD and coregulators?

Other comments

4) Fig. 3A, 7B, and 8A: Why were different media used for the control plates and the fluconazole-containing plates (SC-ura or SC-leu in controls versus YPD in test plates)? Formally, the mutations might affect growth on YPD, not just in the presence of the drug.

5) Line 239: The data shown in Fig. 4B show that the presence of the D1082G mutation in Pdr1 lacking the CRD resulted in a 2-log increase, not a 3-log increase (as stated in the text), in the transformation efficiency.

6) Line 246: “Figure 4B” should probably read “Figure 4C”.

7) Line 288: “compare lanes 3 and 6” should read “compare lanes 2 and 6”. One can also refer to Fig. 5C, in which protein levels were quantified, and compare columns 3 and 5.

8) The legends to Figs. 2 to 8 should state that a representative western blot (out of two, see methods) is shown in each case, and the error bars should be explained (standard deviations cannot be calculated from two values).

9) Line 81: “...overproduction of target genes...” should better read “...overexpression of target genes...”. As it stands, this would be a gene amplification, which is not the case.

10) Lines 565-566: What are two “biologically independent experiments” – two independent experiments (performed on different occasions) or two biological replicates (tested in parallel)?

11) Lines 392-395: “Since PDR1 is an autoregulated gene, it is difficult to ensure that the enhanced transcription of Pdr1-regulated genes is not due to the linked increase in expression of PDR1 itself seen in these GOF mutant strains rather than increased activity of each polypeptide chain”. I have difficulties understanding this line of argumentation. If the GOF mutations did not alter the transcriptional activity of Pdr1, how would the increased expression of the mutated alleles (by autoregulation) be brought about? This potential (but disproven) scenario should be explained, e.g. did the authors hypothesize that the GOF mutations affect PDR1 transcript or Pdr1 protein stability (already excluded in their previous study, see ref. 10)? The sentence also implies that demonstrating increased activity of each Pdr1 molecule dismisses the contribution of the increased Pdr1 levels. However, it is likely that both increased Pdr1 activity (i.e. of each single Pdr1 molecule) and increased levels of the mutated transcription factor contribute to the overexpression of Pdr1 target genes.

Reviewer #2: In this manuscript, Simonicova and colleagues aimed to elucidate how known gain of function (GOF) mutations in PDR1 cause hyperactivity of the protein and Candida glabrata fluconazole resistance. Numerous previous publications have established that GOF mutations in PDR1 lead to increased expression of ATP-binding cassette transporter genes including CDR1. GOF mutations also lead to higher expression of PDR1 through an autoregulatory mechanism, which drives target gene expression (Khakhina, S. et al. Mol Microbiol. 2018. 107:747). In this manuscript, the authors uncouple the effects of the GOF mutations on the levels and activity of Pdr1. Using a strain where PDR1 expression is not controlled by its endogenous promoter and therefore cannot autoregulate, the authors demonstrate that D1082G and R367W GOF mutants induce higher levels of Cdr1, while having lower levels of Pdr1 than the wild-type strain. This demonstrates that Pdr1 GOF mutations increase Pdr1 activity through additional effects beyond altering Pdr1 levels. The authors demonstrate that two different mutations in the transcriptional activation domain (D1082G mutation or LWG1097AAA mutation) substantially increase fluconazole resistance and ameliorate the toxicity of a hyperactive Pdr1 mutant that has its central regulatory domain (CRD) deleted. In addition, the authors show that many of the GOF mutants have effects that are dependent on the known Pdr1 co-activator, Mediator protein Med15A, with the D1082G mutation causing increased occupancy of Med15A at the Cdr1 promoter.

Overall, the writing of this manuscript lacked accuracy and precision, causing many of the authors conclusions to be communicated in ways that are misleading or incorrect. Substantial editing of the manuscript text should be made to strengthen the paper. A limited set of novel mechanistic insights were achieved in this manuscript, but the majority of effects seen were a result of altered Prdr1 levels. Specific comments to improve the manuscript are provided below.

Specific points:

1. Controlling PDR1 expression with the non-endogenous promoter MET3 is sufficient to cause increased resistance to fluconazole (Figure 2A). This is likely due to higher expression of the Pdr1 protein, as demonstrated in Figure 2B and 2C. It would be helpful for the authors to comment on this finding in the text. In addition, the authors could provide insight into why the MET3 promoter increases the Pdr1 protein levels by almost 15-fold but the increase in activity measured by Cdr1 levels is only two-fold.

2. Most of the conclusions in the paper are based on quantifications of western blots, but these experiments could be strengthened significantly by statistical analyses. For example, is the increase in Pdr1 protein levels in the GOF mutants seen in Figure 2B and 2C statistically significant? Are the differences in Cdr1 levels between the triple mutant and the quadruple mutant in Figure 3C (described on lines 218-221) statistically significant? It also seems that error bars are not always visible (ex. Figure 8C, error bars of the Pdr1 levels are only visible for one strain) and there is no description of what the error bars demonstrate (standard deviation or standard error, how many biological replicates). If statistics cannot be done on the western blots, quantitative RT-PCR could be performed to monitor the expression of CDR1.

3. It would be useful to include the data not shown on the protein levels for Prd1 and Cdr1 for the other GOF mutants in the CRD domain or included as a Supplementary figure (line 212-213, Figure 3).

4. Some of the conclusions about Figure 3 are misleading. On lines 202- 204 and 414-415 the authors indicate that the quadruple mutant showed less fluconazole resistance than the D1082G Pdr1 at 10 µg/mL of fluconazole but all of the mutants show a similar resistance to fluconazole in the pdr1Δ strain. The statement is only true in the pdr1Δ med15AΔ strain. On lines 204-205 and lines 410-411 the authors indicate that the triple mutant is the strongest GOF allele, but again this is only true in the pdr1Δ med15AΔ strain. These statements should be rewritten for accuracy.

5. In the discussion of Figure 5A, the authors do not comment on the intriguing finding that the D1082G mutation ameliorates the toxicity of the Δ255-968 PDR1 when the wild-type PDR1 is repressed with met. This is confounded by the fact that the D1082G mutant causes lower levels of Pdr1 than the Δ255-968 PDR1 alone (also seen in Figure 6). The authors should comment on why the D1082G would affect Δ255-968 Pdr1 protein levels.

6. The authors indicate that the D1082G mutation reduces the function of the Δ255-968 PDR1, as it is less effective at supporting the fluconazole resistance (lines 304-307). This seems misleading, as the difference between the two mutants is very minor and is only seen in the absence of Med15A. At most, this could be interpreted as there being a minor increase in dependence on Med15A. The authors also conclude that the D1082G mutation reduces the capacity of the Δ255-968 PDR1 to induce Cdr1 levels (lines 313-316, 440-442), which again is only seen in the absence of Med15A and is confounded by the decrease in Pdr1 levels seen. These conclusions should be softened and the limitations addressed.

7. In Figure 7C and D, the authors compare the levels of Pdr1 and Cdr1 across 5 different western blots. It is not appropriate to compare quantification of western blot bands that were not run at the same time, on the same gel, or have been spliced and therefore we cannot interpret the results. These western blots and quantification should be repeated appropriately.

8. In lines 375-377, the authors describe the double mutant as being highly defective in expression of both Pdr1 and Cdr1. This is not true, as Pdr1 levels in the double mutant are similar to the wild-type strain or the AAA single mutant. The double mutant has a substantial decrease in Cdr1 levels compared to the D1082G mutant and a modest decrease compared to the AAA mutant. The wording of this statement should be corrected.

9. In many instances in the paper, the authors imply novel findings that Pdr1 interacts with multiple co-activators. For example, lines 116 to 118 implies that the authors provide genetic evidence that identifies a suite of coactivators that interact with Pdr1, which is not true. The authors state in the abstract that they provide genetic evidence for an element within the transactivation domain (TAD) that mediates the interaction of Pdr1 with coactivators. This is misleading, as it was only found that the D1082G mutation affects Med15 recruitment to the CDR1 promoter. These findings should be written explicitly.

10. The manuscript could be substantially strengthened by efforts to assess the effect of the genetic alterations on interactions with the other known coactivators Bre5 and Jjj1 or identifying new coactivators.

***11. The paper could be strengthened if the authors showed the effects of alterations in Pdr1 on the expression of additional Pdr1 target genes besides CDR1 that are known to affect fluconazole efflux, such as CDR2 and SNQ2. This is integral, as previous studies have found that some GOF mutations do not alter expression of CDR1, but cause upregulation of SNQ2 (Torelli, R. et al. Mol. Microbiol. 2008. 68:186). Alternatively, the authors could perform functional assays to monitor the effect on drug efflux.

12. It is appreciated that the system is very complex and multifaceted, but a more thorough explanation of different models of how the GOF mutations could be affecting Pdr1 activity should be presented. How do the authors explain the finding that the three single GOF mutations in the CRD are Med15A dependent but the triple mutant is not? A discussion should be provided to explain how the repression of the CRD occurs via interactions between the TAD, as suggested in the figure legend, but is still affected by Med15A. Further discussion should be provided about the connection between the D1082G mutant and the LWG1097AAA mutant. For example, why is the increased association of Med15A with the CDR1 promoter seen in the D1082G mutant lost upon the additional LWG1097AAA mutation.

13. The discussion section of the paper is lacking any discussion of the implications of the findings or how they expand on findings in the field. The authors could include a discussion of the known effects of Pdr1 GOF mutations on virulence and fitness of C. glabrata in animal models (Ferrari, S. PLoS Pathogens. 2009. 5:e1000268).

Minor points

- Line 32 – amino acids.

- Line 45 – unclear wording “define nonidentical negative inputs”, since GOF mutations give a gain of function and not a negative input.

- Line 91 – unclear wording for “is a toxic transactivator”.

- Lines 99-101 – name the two transactivating negative regulators, Bem1 and Jij1.

- Line 148 – D1082G.

- Line 168 – what is it meant by “specific activity”.

- Line 180 – Remove the and.

- The legend is missing in Figure 3C.

- Lines 217-218 – It seems misleading to say that the levels of Pdr1 were similar across all mutants when there is almost a two-fold difference in levels between the D1082G mutant and the Triple mutant.

- Lines 233-235 – include a reference to Figure 4A, which is currently missing.

- Line 239 – it appears to be a 2-log difference not a 3-log difference, as described.

- The reference to Figure 4B should be removed from line 246, since the figure does not contain data with the strain of the Δ255-968 mutant of Pdr1 with a chromosomal copy of PDR1.

- Figure 4 D and E would be easier to interpret if the western and the bar graphs were plotted with the strains in the same order.

- Lines 270-272 – the description of the TAP tag should be moved to Figure 4 where the TAP tagged mutants are first used.

- Lines 286-287 (and various places) – Unclear wording of internally deleted Pdr1. Refer to it by the domain deletion or something more descriptive.

- Line 288 – the relevant comparison should be 2 and 6, not 3 and 6.

- Line 339-340 – the lower level of drug resistance of the LWG1097AAA mutation compared to the Δ255-968 Pdr1 alone can only be assessed in the absence of Med15A. This should be clarified.

- A reference to Figure 8D needs to be added somewhere in the text.

- Line 423 – lesion should be changed to mutation.

- Line 489 – C. glabrata should be italicized.

- Line 775 – amino acids.

- The references should be formatted properly so that all species names are italicized.

**Have all data underlying the figures and results presented in the manuscript been provided?**

Reviewer #1: Yes

Reviewer #2: Yes

PLOS authors have the option to publish the peer review history of their article (what does this mean?). If published, this will include your full peer review and any attached files.

Reviewer #1: No

Reviewer #2: No

---

## [Decision Letter · Decision Letter 1]

14 Jul 2020

Dear Dr Moye-Rowley,

Thank you very much for submitting your Research Article entitled 'Functional information from clinically-derived drug resistant forms of the Candida glabrata Pdr1 transcription factor' to PLOS Genetics. Your manuscript was fully evaluated at the editorial level and by independent peer reviewers. The reviewers appreciated the attention to an important topic but identified some aspects of the manuscript that should be improved.

The minor issues that remain (see reviews below) are:

1.  Interpretation of the data in Figure 3 - softening the conclusion in light of the reviewer's comments seems appropriate.

2.  Possible inconsistency of Figure 5B and text line 304 - please make sure that the experiment and reagents are described accurately and consistently.

3.  Some minor editorial issues, as pointed out by the reviewers.

We therefore ask you to modify the manuscript according to the review recommendations before we can consider your manuscript for acceptance. Your revisions should address the specific points made by each reviewer.

[LINK]

Yours sincerely,

Aaron P. Mitchell, PhD

Guest Editor

PLOS Genetics

Gregory P. Copenhaver

Editor-in-Chief

PLOS Genetics

Reviewer's Responses to Questions

**Comments to the Authors:**

Reviewer #1: The authors have discussed and addressed my previous comments in their response letter and revised manuscript (I had not suggested additional experiments). I still think that my explanation for the fact that activating mutations in the TAD decrease the activity of Pdr1 forms that are already hyperactivated by a CRD mutation is simpler and straightforward (previous comment 1; I had compared this with resistance mutations in Erg11 as an analogy for ease of understanding, but this was not meant as a mechanistic comparison of enzymes and transcription factors). However, this is up to the authors, and readers can interpret the data for themselves.

I found some minor errors that should be corrected:

1) Line 23: “Figure 3B and C” should read “Figure 3C and D”

2) There appears to be an error in Fig. 5B and the corresponding text (line 304) that had escaped me in the original version. If I understood the experiment correctly, an antibody against wild-type Pdr1 was used here, not an anti-TAP antibody, because the latter would not detect the longer wild-type Pdr1 produced by PDR1 expression from the MET3 promoter. This is correctly described in the legend to Fig. 5B, but the labeling of the corresponding band in Fig. 5B is also wrong (delete TAP). Or did the wild-type PDR1 expressed from the MET3 promoter also contain the TAP tag (see lines 288-292)? In that case, “MET3-PDR1” should read “MET3-TAP-PDR1” in the upper left part of Fig. 5B (the use or not of italics in the designations was confusing to me) and the description in the legend be corrected. The authors should probably check the whole manuscript for the correct description of the use of anti-Pdr1 or anti-TAP antibodies.

Reviewer #2: In this revised version of the manuscript, Simonicova and Moye-Rowley have substantially improved their submission which characterizes the effect of Gain of Function mutations in the transcription factor Pdr1 on Candida glabrata fluconazole resistance.

The authors have dramatically improved the writing and clarity of the text. They have added additional information where necessary and have added additional experimental data as recommended, including the RT-PCR of CDR1 levels for the additional GOF mutants and the RT-PCR of other PDR1 target genes.

While the majority of major concerns have been addressed, some issues still remain. The biggest issue is with the interpretation of the data in Figure 3. The authors conclude that the introduction of the D1082G allele into the Triple mutant (to generate the Quadruple mutant) decreases the transcriptional activity of the strain (lines 438-439). They claim on Lines 237 to 238 that “the Triple mutant drove the highest level of Cdr1 expression, an effect that was reversed by introduction of the D1082G allele in the Quadruple mutant.”

This reviewer does not agree with these conclusions. While there is a 50% decrease in the expression levels of CDR1 by RT-PCR in the Quadruple mutant versus the Triple mutant, this is only seen in the absence of Med15A. Additionally, this 50% decrease in CDR1 levels is not reflected at the protein level in the absence of Med15A (in Figure 3C and D), suggesting that it is unlikely that these small transcriptional differences are causing the increased fluconazole sensitivity of the Quadruple mutant observed in A.

In the presence of Med15A, the addition of the D1082G mutation has no effect on the CDR1 levels compared to the Triple mutant. While there is a slight difference in Cdr1 protein levels in the Western blot between the Triple mutant and Quadruple mutant in Figure 3C and D, this effect appears minor and with the high levels of error and lack of statistical significance testing, this reviewer does not feel confident in that conclusion either. It is very clear that adding the D1082G mutation to the Triple mutant increases the fluconazole sensitivity of C. glabrata in the absence of Med15A, which the authors can emphasize in the text, but it seems misleading to conclude that this is due to changes in CDR1.

This reviewer appreciates that they tried to strengthen their point by including the quantitative RT-PCR data for the three other Pdr1 target genes. As the level of expression for the additional target genes is low and they do not show the same exact patterns as the CDR1 levels, they do not help to strengthen the authors’ claims on this point.

There are additional minor Comments to improve the manuscript:

- Line 32- “amino acids”

- In Figure 3B caption, it would be useful to mention that cDNA synthesis was performed and DNA is normalized to TEF1

- Line 230 – should refer to Figure 3C and D

- Lines 232-233 – It is unclear why the authors are focusing on the fact that in the R376W Cdr1 expression is highly dependent on the presence of Med15A, when the expression of Cdr1 is highly dependent on Med15A in all the mu as noted in the text.

- Line 275 – It should be written more clearly what the Cdr1 levels are being compared to. Is it a comparison of in the presence of PDR1 versus the absence of PDR1 for the D1082G mutant or is in the comparison of the D1082G mutant compared to the delta255-968 mutant?

- Line 276 – The authors could say in Figure 4C instead of above.

- Lines 333 -335 - The authors should acknowledge that the levels of Pdr1 decrease in Figure 6B when D1082G is added to the Δ255-968 mutant, as it could be contributing to the decrease in Cdr1 levels.

- Line 337 – This is only true in the absence of Med15A.

- Lines 362-366 – The strain has 3 modifications to Pdr1, not 2, as it is also a deletion of 255-968.

- Figure 7D – add a label for the Cdr1 quantification graph

- Lines 417-419 – The authors should clarify that they are referring to the D1082G and R376W mutants and the data is shown in Figure 2. In all areas of the discussion, it would increase clarity if the authors would mention which figures they are referring to.

Reviewer #3: The authors undertook laudable efforts to address all points raised in my report on the original submission. Further, the authors added new experimental data and introduced extensive revisions, most of which were requested by the other reviewers.

While I still disagree with the response to the minor point A, it is acceptable to leave Figure 1 in the main manuscript, given that it adheres to the author guidelines of the journal.

Overall, this is a nice piece of work, offering new and interesting mechanistic insights about the function of the Pdr1 transcription factor and its cross-talk with Med15. This reviewer therefore looks forward to seeing this manuscript published in PLoS Genetics.

**Have all data underlying the figures and results presented in the manuscript been provided?**

Reviewer #1: Yes

Reviewer #2: Yes

Reviewer #3: Yes

PLOS authors have the option to publish the peer review history of their article (what does this mean?). If published, this will include your full peer review and any attached files.

Reviewer #1: No

Reviewer #2: No

Reviewer #3: No

---

## [Editor Report · Decision Letter 2]

22 Jul 2020

Dear Dr Moye-Rowley,

We are pleased to inform you that your manuscript entitled "Functional information from clinically-derived drug resistant forms of the Candida glabrata Pdr1 transcription factor" has been editorially accepted for publication in PLOS Genetics. Congratulations!

Yours sincerely,

Aaron P. Mitchell, PhD

Guest Editor

PLOS Genetics

Gregory P. Copenhaver

Editor-in-Chief

PLOS Genetics

Comments from the reviewers (if applicable):

**Data Deposition**

http://datadryad.org/submit?journalID=pgenetics&manu=PGENETICS-D-20-00307R2

**Press Queries**

---

## [Editor Report · Acceptance letter]

19 Aug 2020

PGENETICS-D-20-00307R2 

Functional information from clinically-derived drug resistant forms of the Candida glabrata Pdr1 transcription factor 

Dear Dr Moye-Rowley, 

We are pleased to inform you that your manuscript entitled "Functional information from clinically-derived drug resistant forms of the Candida glabrata Pdr1 transcription factor" has been formally accepted for publication in PLOS Genetics! Your manuscript is now with our production department and you will be notified of the publication date in due course.

With kind regards,

Jason Norris

PLOS Genetics

On behalf of:
